

# Quantifying the impact of synoptic weather types, patterns, and trends on energy fluxes of a marginal snowpack

Andrew Schwartz[1], Hamish McGowan[1], Alison Theobald[2], Nik Callow[3]

[1]Atmospheric Observations Research Group, University of Queensland, Brisbane, 4072, Australia
[2]Department of Environment and Science, Queensland Government, Brisbane, 4072, Australia
[3]School of Agriculture and Environment, University of Western Australia, Perth, 6009, Australia

*Correspondence to:* Andrew J. Schwartz (Andrew.Schwartz@uq.edu.au)

**Abstract.**

Synoptic weather patterns and teleconnection relationships across a 39 year climatology are investigated for their impact on energy fluxes driving ablation of a marginal snowpack in the Snowy Mountains, southeast Australia. K-means clustering applied to ECMWF ERA-Interim data identified common synoptic types and patterns that were then associated with in-situ snowpack energy flux measurements. The analysis showed that the largest contribution of energy to the snowpack occurred immediately prior to the passage of cold fronts through increased sensible heat flux as a result of warm air advection (WAA) ahead of the front. Indian Ocean Dipole and Southern Oscillation Index phase combination had a strong relationship with energy flux, with eight of the ten highest annual snowpack energy fluxes occurring during a negative IOD phase and positive SOI phase. Overall, seasonal snowpack energy flux over the 39 year period had a decreasing trend that is likely due to a reduction in the number of precipitation generating cold fronts and associated preceding WAA ahead of precipitation. This research is an important step towards understanding changes in surface energy flux as a result of shifts to the global atmospheric circulation as anthropogenic climate change continues.

## 1    Introduction

### 1.1    Synoptic weather influences on snow and glacier processes

Water generated in mountainous regions is a commodity that over 50% of the world's population depends on for daily life (Beniston, 2003). Arguably, the most important role in the generation and regulation of these water resources is that of montane snowpacks. These have been referred to as "water towers" (Viviroli et al., 2007) due to their capabilities for storage and slow releases of meltwater. Many snowpacks are undergoing reductions in spatial and temporal extent as a result of anthropogenic climate change (Pachauri et al., 2014). Understanding the physical drivers of snowpack ablation, including synoptic-scale influences, is critical to help assess future water resource availability in mountainous regions as climate change continues.

Snowfall has been related to synoptic weather types in numerous studies globally including in Athens (Prezerakos and Angouridakis, 1984), the central and eastern United States (Goree and Younkin, 1966), the Tibetan Plateau (Ueno, 2005), Budapest (Bednorz, 2008a), and the central European lowlands (Bednorz, 2011). However, work on relationships between snowmelt and synoptic weather types is relatively scarce. Bednorz (2008b) identified increased air temperature and rain-on-snow events as causes for rapid snowmelt ($> 5$ cm day$^{-1}$) in the Polish-German lowlands as a result of west-southwest airflows over Central Europe during positive phases of the North Atlantic Oscillation (NAO). Similar work has been conducted in North America by Grundstein and Leathers (1998) who were able to identify three main synoptic weather types responsible for significant snowmelt events on the northern Great Plains, all of which included cyclonic influence with different low pressure centre locations



and warm air advection to the region. While some knowledge exists on synoptic drivers of snowpack ablation,
further research is needed to understand synoptic effects on ablation processes over snowpacks with varying
characteristics. Furthermore, research related to synoptic influences on the surface energy balance over marginal
snowpacks are rare. Hay and Fitzharris (1988) studied the influence of different synoptic weather types on glacier
ablation and snowpack melt, while Neale and Fitzharris (1997) used surface energy flux measurements to
determine which synoptic types resulted in highest ablation in the Southern Alps, New Zealand. These studies
found net radiation was the dominant term in ablation, but also noted that the contributions made by each term
varied largely depending on the synoptic type and its meteorology. A common characteristic between these studies
and others in various regions is that they focused primarily on the surface meteorology for synoptic classifications
rather than multiple level analysis, which enables insight to the potential influence of mid and upper-level
atmospheric conditions on surface – atmosphere energy exchanges. Regardless, no analysis at any level exists on
synoptic type influence on snowpack ablation within Australia.

## 1.2 Synoptic weather types and trends in the Australian Alps

Precipitation in the Australian Alps is crucial to agriculture, the generation of hydroelectric energy, and recreation
and was estimated to be worth $9.6 billion per year in 2005 (Worboys and Good, 2011). A maximum in
precipitation in the Snowy Mountains typically occurs during the cooler months of June-September when it falls
as snow at elevations above 1400 m and accounts for twice as much precipitation as during the warmer periods
of the year (Chubb et al., 2011). While the snowpack of the Snowy Mountains typically exists for relatively short
periods compared to those of other regions where winter temperatures are lower and higher snowfall amounts
occur such as the European Alps and Rocky Mountains, USA, it is still a vital resource for southeast Australia
(SEA).

Synoptic weather types in Australia have been changing in recent decades in response to the impact of climate
change on background climate states (Hope et al., 2006; Theobald et al., 2016). For example, increases in daily
maximum temperature and reductions in precipitation during autumn and winter have been noted in SEA as a
result of anomalously high surface pressure during positive periods of the Southern Annular Mode (SAM)
(Hendon et al., 2007). Cai et al. (2005) also showed an increase in SAM value as a response to all global warming
experiments using the CSIRO Mark 3 climate model indicating a further poleward shift in the location of synoptic
systems. However, it has been suggested that the SAM accounts for a relatively small portion of seasonal rainfall
variability in Australia and other larger impacts on synoptic weather from other sources are likely (Meneghini et
al., 2007).

Rainfall in SEA is known to be influenced by the Southern Oscillation Index (SOI) and Indian Ocean Dipole
(IOD). Stone and Auliciems (1992) showed above (below) average rainfall amounts correspond to high (low) SOI
values through analysis using the k-means clustering technique. However, Pepler et al. (2014) identified IOD as
the dominant control on cool-season rainfall in SEA when compared to the El Nino Southern Oscillation (ENSO)
due to its effect on westerly wind patterns. In addition, positive IOD phases and associated shifts in synoptic types
have been identified as precursors to large bushfires in Austral spring and summer such as the Black Saturday and
Ash Wednesday events, two of the most severe fires in Australia's history, due to reductions in rainfall and soil
moisture (Cai et al., 2009a). Comparisons of the effects of IOD and ENSO phases on precipitation in SEA by
Ummenhofer et al. (2009) showed the positive phase of the IOD as the primary driver of large droughts as a result



of reduced moisture advection from the tropical seas northwest of Australia. The relationship between positive
IOD phase and reduction in rainfall for SEA is particularly concerning when climatological trends have noted an
increased likelihood of positive IOD periods as a result of climate change (Cai et al., 2009b).

### 1.3    The Australian snowpack

Characteristics of the snowpack in the Australian Alps have been examined in a number of studies with focus on
spatial and temporal snow cover variability (Budin, 1985; Duus, 1992), influence on catchment hydrology (Costin
and Gay, 1961), the energetics of snowpack melt (Bilish et al., 2018), and isotopic composition of precipitation
(Callow et al., 2014). Given observed declines in snow cover, climate change has become a central focus of this
research (Chubb et al., 2011; Hennessy et al., 2008; Nicholls, 2005; Reinfelds et al., 2014; Whetton et al., 1996)
as any changes to energy flux over the region will significantly impact the already marginal snowpack. Hennessy
et al. (2008) showed that future projections for the Australian snowpack predict reductions in annual areal snow
cover of 10-39% by 2020 and 22-85% by 2050. Observations indicate that reduction in snow cover is already
occurring with shortened annual periods of wintertime precipitation. Nicholls (2005) found reductions of 10%
and 40% in the maximum snow depth and snow depth at the first October measurement respectively from 1962
to 2002. In addition, wintertime precipitation was shown to have reduced by an average of 43% in high elevation
regions from 1990-2009 (Chubb et al., 2011), though much of this could have been due to several severe droughts
that occurred during the study period. Fiddes et al. (2015) showed that snowfall, snow accumulation, and snow
depth were highly correlated with temperature and that warming, as a result of climate change, could lead to
further reductions in the SEA snowpack. The importance of the water generated in the Australian Alps, reduction
in wintertime precipitation amounts and frequency, and high spatiotemporal variability of snow accumulation and
ablation (Budin, 1985) warrants an understanding of the energetics of Australia's snowpack as they pertain to the
influences of shifting synoptic-scale circulations.
Significant work has been conducted on identification of patterns and trends in Australian synoptic climatology
as it pertains to precipitation variability (Chubb et al., 2011; Pook et al., 2006; 2010; 2012, 2014; Theobald et al.,
2016). However, impacts on surface energy fluxes as a result of synoptic types and changing climatological
conditions have not been explored as they have in other regions. The objective of this study is to identify the
synoptic weather types that contribute the highest amounts of energy fluxes to the Australian snowpack, and their
climatological trends. This is accomplished through: 1) the identification and classification of common synoptic
types during periods of homogeneous snow cover, 2) attribution of snowpack energy flux characteristics to each
synoptic type, 3) construction of energy balance patterns as they pertain to common synoptic
patterns/progressions, and 4) investigation of relationships between trends in snowpack energy flux and
teleconnections.

## 2    Methods

### 2.1    Study site and climate

Energy flux measurements were made 16 km west of Lake Jindabyne at the Pipers Creek catchment headwaters
(36.417°S, 148.422°E) at an elevation of 1828 m in the Snowy Mountains, Kosciuszko National Park, New South
Wales (NSW), Australia (Figure 1). The catchment is classified as sub-alpine and contains grasslands, sub-alpine
bogs, and sub-alpine woodland (Gellie, 2005). The surrounding areas contain a mixture of living and dead
*Eucalyptus pauciflora* (Snow Gum) trees and open grassland areas with fens and alpine bogs. Many of the Snow



Gums were impacted by fire in 2003, and have experienced slow regrowth. The area's mixed characteristics of
forested and open grasslands with alpine wetlands within the Pipers Creek study catchment and immediately
surrounding the flux tower site are representative of those found throughout the Australian Alps.
The Snowy Mountains are characterized by relatively mild weather conditions compared to other mountain
ranges. Winter temperatures are typically around 0°C with mean low temperatures during July (the coldest month)
at -5°C and mean high temperatures between 2-4°C (Bureau of Meteorology, 2018b) that readily allow for melt
of the snowpack. Precipitation reaches a peak during the winter and spring months with the majority falling as
snow during July and August before ablating completely in late spring or early summer. Prevailing winds in the
region are westerly and reach a maximum in intensity during September and October, but intense wind events are
common throughout winter and spring during the passage of cold fronts. Snowpack properties in the catchment
are consistent with those of maritime snowpacks that are associated with basal melting, high temperatures, and
high wind speeds (Bilish et al., 2018; Sturm et al., 1995).

### 2.2    Instrumentation

An energy balance site (Figure 2) was installed in the Pipers Creek catchment headwaters on June 10th of 2016.
The site consists of a Campbell Scientific eddy covariance system to measure fluxes of latent and sensible heat at
10 Hz at a height of 3.0 m above ground level (AGL). A Kipp and Zonen CNR4 radiometer (3.0 m AGL) was
used to measure incoming and outgoing shortwave and longwave radiation to allow for comparisons of all
radiation components rather than simply net all-wave radiation. Ambient air temperature and relative humidity
were measured at the top of the mast by a Vaisala HMP155 probe at ~3.1 m above ground level. A Hukseflux
heat flux plate measured ground heat flux at a depth of 5 cm and was placed approximately 0.5 m from the centre
of the mast to minimize any influence the mast could have on snow accumulation above the sensor. Surface
temperatures were monitored using an Apogee Instruments SI-111 infrared radiometer at approximately 2 m from
the centre of the mast. Details on the instruments used for each measurement are shown in Table 1.
The eddy covariance system was controlled by a Campbell Scientific CR3000 micrologger with raw 10 Hz data
stored on memory cards at the site to allow for in-depth analysis of data as a back-up to the 30-minute averages
that were also recorded. In addition, data tables were recorded that contained notes for all measurements to allow
for potential identification of problematic measurements in real-time and analysis. All data was monitored in near-
real time through download of telemeted data with the exception of the 10 Hz tables that were collected through
replacement of the micrologger memory cards approximately every four months.
Precipitation data from an ETI Instrument Systems NOAH II weighing gauge located approximately 1 km to the
northwest of the energy balance site at elevation of 1761 m was supplied by Snowy Hydro Limited (SHL). A 6 m
diameter DFIR shield was used around the gauge in order to prevent wind-related under-catch of snowfall
(Rasmussen et al., 2012), and was additionally sheltered by vegetation to the west.

### 2.3    Identification of snow cover periods

Homogeneous snow cover is crucial to accurate measurement and analysis of snowpack energy balance (Reba et
al., 2009). Periods with homogeneous snow cover were determined using data from the Pipers Creek
instrumentation site and were cross referenced to manual snow measurements made at the Spencers Creek Snow
Course 6.6 km northwest of the Pipers Creek field site (Snowy Hydro Ltd, 2018). Periods with surface





temperatures above 1.5°C as measured by the SI-111 infrared radiometer that did not correspond to rain-on-snow
events and periods with albedo measurements less than 0.40 (Robock, 1980) were considered not to have
heterogeneous snow cover and were eliminated.

**2.4     Synoptic classification of snow cover days**

Synoptic weather type classification of homogeneous snow cover days was conducted using synoptic typing
methods adapted from Theobald et al. (2015). European Centre for Medium-Range Weather Forecasts (ECMWF)
ERA-Interim reanalysis data (Dee et al., 2011) with a 0.75° X 0.75° resolution was obtained for each day from
June 10th, 2016 through October 31st, 2017. This date range was chosen to ensure inclusion of all potential dates
with snow cover during the 2016 and 2017 snow seasons after the initial instrument tower installation on June
10th, 2016. Variables included in the reanalysis data consisted of mean daily values of Mean Sea Level Pressure
(MSLP); temperature and relative humidity values at 850, 700, 500, and 250 hPa; wind vectors at 10 m AGL,
850, 700, 500, and 250hPa; and 1000-500 hPa geopotential heights. The domain of the included variables was
limited to 20°S - 46°S and 120°E -160°E, ensuring coverage of synoptic scale systems affecting the Australian
Alps.
Focus was placed on analysis of temperature ($T_d$) and relative humidity ($RH$) values because of their impact on
latent heat, sensible heat, and radiative fluxes (Allan et al., 1999; Reba et al., 2009; Ruckstuhl et al., 2007; Webb
et al., 1993). Relative humidity values at 850, 700, and 500 hPa were used to investigate the potential influence
of cloud cover. MSLP and wind vector analysis at the 850, 700, 500, and 250 hPa levels allowed for the
identification of $T_d$ and $RH$ advection (Pook et al., 2006) into the Australian Alps. Thickness between 1000-500
hPa was used to determine frontal positions relative to the Australian Alps (Pook et al., 2006) and accordingly the
Pipers Creek field site.
Days within the ERA-Interim data that matched snow cover days were extracted and analysed using the k-means
clustering algorithm developed by Theobald et al. (2015). The algorithm was tested for 1-20 clusters and an elbow
plot of the cluster distances was used to identify the optimum number of clusters (Theobald et al., 2015), which
was seven. The identification of an elbow in the plot (Figure 3) at seven clusters indicates a reduction to the benefit
of adding additional clusters as the sum of distances for additional clusters fails to yield significant reductions
beyond that point (Wilks, 2011).
Clustering of the synoptic conditions for each day was verified through manual analysis of MSLP and 500 hPa
charts from the Australian Bureau of Meteorology (BOM) (Bureau of Meteorology, 2018). Cloud cover for each
type   was   investigated   and   verified   through   the   use   of   visible   band   Himawari-8   satellite   data
(https://www.ncdc.noaa.gov/gibbs/) at 03:00 UTC (13:00 local time) with one of three categories assigned to each
day studied; 1) no cloud cover, 2) partial cloud cover, or 3) complete cloud cover. Cloud cover was investigated
at midday to avoid misclassification due to short-lived clouds that appear over the area during the dawn and dusk
periods.
Manual verification of the k-means clustering algorithm using BOM synoptic charts identified four days (2.33%)
out of the 172 classified during the 2016 and 2017 seasons that had been classified incorrectly and they were
subsequently moved to their correct synoptic type. Three of the four misclassified days were early (June 7th, 2016)
or late (September 19th and 22nd, 2016) in the snowpack seasons with the fourth occurring in the middle of winter





on July 31st of 2017. Synoptic characteristics from these days tended to be complicated with no discernible
dominant features that matched those of classified types. This is likely due to shifting synoptic conditions between
seasons related to poleward or equatorial shifts in westerly winds.
**2.5    Snowpack energy accounting**
Accurate measurement of snowpack energy balance and associated melt can be difficult due to snowpack
heterogeneity (Reba et al., 2009) and problems with energy balance closure (Helgason and Pomeroy, 2012). The
snowpack energy balance can be expressed as:
$Q_m = Q^* + Q_h + Q_e + Q_g + Q_r$ (1)
where the energy available for snow melt ($Q_m$) is equal to the sum of net radiation exchange ($Q^*$), sensible ($Q_h$)
and latent ($Q_e$) heat flux, ground heat flux ($Q_g$), and the energy flux to the snowpack from liquid precipitation
($Q_r$) (Male and Granger, 1981; McKay and Thurtell, 1978).
While net all-wave radiation exchange ($Q^*$) is used for basic analysis of the snowpack energy balance, a
decomposition into its individual components is necessary to understand the role of short and longwave radiation
exchange in snowpack energetics (Bilish et al., 2018). Therefore, net radiation should be broken into its net flux
terms:
$Q^* = K^* + L^*$ (2)
that quantify the net shortwave ($K^*$) and net longwave ($L^*$) components.
The approach taken within this paper is to examine net radiative flux components individually, similar to the
methods used by Bilish et al. (2018), to be precise in the identification of synoptic-scale effects on snowpack
energy fluxes through differences in temperature, relative humidity, cloud cover. $Q_m$ calculation and comparisons
of snowpack energy flux terms were performed using the terms in Eq. (1), but with the net radiation terms ($K^*$ and
$L^*$) used rather than summed as net all-wave radiation ($Q^*$) only.
**2.6    Energy flux measurements of synoptic types**
Energy flux measurements made at the Pipers Creek tower underwent a series of measurement corrections prior
to analysis. Coordinate rotation based on the methods of Wilczak et al. (2001) was applied on the 10 Hz EC data
to remove levelling errors in sonic anemometer mounting when calculating fluxes. In addition, frequency
corrections were made to the EC data to account for sensor response delay, volume averaging, and the separation
distance of the sonic anemometer and gas analyser when calculating fluxes (Campbell Scientific, 2018b). Finally,
WPL air density corrections (Webb et al., 1980) were made by the software to account for vertical velocities that
exist as a result of changing air mass density through fluxes of heat and water vapour.
$Q_h$ and $Q_e$ flux were calculated using the EC equations of:
$Q_h = -\rho C_p (\overline{w'\theta'})$ (3)
$Q_e = -\rho L_v (\overline{w'q'})$ (4)



where $\rho$ is air density (kg m$^{-3}$), $C_p$ is the specific heat of air (1005 J kg$^{-1}$ deg$^{-1}$), $\overline{w'\theta'}$ is the average covariance
between the vertical wind velocity $w$ (ms$^{-1}$) and potential temperature $\theta$ ($K$), $L_v$ is the latent heat of sublimation
or vaporization of water (J kg$^{-1}$), and $\overline{w'q'}$ is the average covariance between the vertical wind velocity $w$ (ms$^{-1}$)
and specific humidity $q$ (kg kg$^{-1}$) (Reba et al., 2009).
The calculation of energy flux imparted to the snowpack from rainfall ($Q_r$) followed Bilish et al. (2018) and was
determined using three separate calculations to establish approximate wet bulb temperature ($T_w$) (Stull, 2011), the
fraction of precipitation falling as rain ($1 - P_{snow}$) (Michelson, 2004), and total rain flux ($Q_r$) based on
precipitation accumulation over a 30-minute period.

### 2.6.1    Energy flux data quality control

Data quality control was performed on the filtered energy flux data to remove potential erroneous data. Periods
were omitted from analysis where latent and/or sensible heat flux measurements were rated $\geq 7$ by the Campbell
Scientific EasyFlux™ software indicating problems with non-stationarity of wind flow, turbulence characteristics,
or horizontal orientation of the sonic anemometer (Foken et al., 2012). $Q_e$ and $Q_h$ values were also removed when
water vapour signal strength from the gas analyser was $< 0.70$ in order to remove erroneous readings during
periods of precipitation (Campbell Scientific, 2018a; Gray et al., 2018). De-spiking was performed on sensible
and latent heat data by visual inspection and through the application of filtering techniques to remove erroneously
high or low values. Latent heat and sensible heat flux thresholds of -100 Wm$^{-2}$ to 500 Wm$^{-2}$ and -200 Wm$^{-2}$ to 500
Wm$^{-2}$, respectively, were applied to remove erroneous values that had been identified during visual inspection. In
addition, a seven point moving-median filter was implemented over three iterations to remove values more than
3.0 standard deviations away from the median values.
Pre-existing gaps and gaps introduced into the data by the quality control procedures were filled using two
methods described by (Falge et al., 2001a; 2001b). Linear interpolation of missing $Q_e$ and $Q_h$ values was used for
gaps up to 90 minutes in length. Look-up tables using six categories of stability ($\zeta$), five categories of wind speed
($u$), five categories of relative humidity ($RH$), and six categories of the difference between surface and
atmospheric temperatures ($dT$) were used to fill any gaps longer than 90 minutes. Traditionally, mean diurnal
variation values are also used for gap filling procedures (Bilish et al., 2018; Falge et al., 2001a; 2001b). However,
it was determined that using mean values would likely obscure any unique energy balance characteristics of the
synoptic types being investigated and, therefore, was not included as a gap-fill strategy for the data.
Following quality control procedures, 3864 of the initial 12147 records (31.8%) remained in the $Q_e$ data and 4540
records (37.4%) remained in the $Q_h$ data. Linear interpolation yielded an addition of 1693 $Q_e$ values (13.9%) and
1513 $Q_h$ values (12.5%). Look-up tables were the largest source of gap-filled data with the contribution of an
additional 6480 $Q_e$ values (53.3%) and 5991 $Q_h$ values (49.3%). Root Mean Squared Error (RMSE) and Mean
Bias Error (MBE) were calculated for 100 iterations of the gap-fill process using values available following quality
control procedures. This resulted in mean RMSE values of 20.9 Wm$^{-2}$ and 33.9 Wm$^{-2}$ during the observation
period for $Q_e$ and $Q_h$, respectively. Missing values still existed in the $Q_e$ and $Q_h$ data following the gap-filling
procedures, however, these values were less than 1% of the initial 12147 records.



This research uses the energy flux convention where positive values are flux to the snowpack and negative values
are flux away from the snowpack.

**2.7    Determining climatology of snowpack energy flux**

The Snowy Mountains snowpack at elevations similar to Pipers Creek (1828 m) generally exists from June
through the end of October though it may begin as early as late-April and exist into early December (Snowy Hydro
Ltd, 2018). However, for climatological comparisons of synoptic types and snowpack energy flux the period of
most-likely snow cover, June through October, will be considered as the snowpack/snow cover season as times
outside of this period are not consistent in their snow cover properties. This season has been used to minimize
error introduced into the analysis through the incorporation of periods without snow cover.
Following the identification of the optimum number of synoptic types and their corresponding energy flux
characteristics, each day between June 1$^{st}$ and October 31$^{st}$ from 1979 through 2017 (39 seasons) was classified
using the synoptic types developed from the 2016 and 2017 snow cover days. Each type was analysed for trends
in frequency over the 39 year period to develop a climatology of the synoptic types in the Snowy Mountains
region. Annual counts of each type were then determined and were multiplied by the mean daily net snowpack
energy flux for each synoptic type found during the 2016-2017 seasons. Mean daily flux values were then summed
for each year and trends in snowpack energy flux as it pertains to synoptic scale atmospheric conditions were
identified over the 39 year period. We acknowledge our results represent an estimate of the snowpack energy flux
over the 39 year record based on similar synoptic conditions observed during the observational campaign in 2016
and 2017.
Investigation into teleconnection impacts on synoptic types was conducted to determine impacts of the IOD, SAM,
and SOI phase on synoptic type occurrence. Monthly data for each of the teleconnections from 1979 to 2017 was
obtained from the National Oceanic and Atmospheric Administration's (NOAA) Earth System Research
Laboratory (ESRL) (https://www.esrl.noaa.gov/psd/data/climateindices/list/). Mean values for each synoptic type
were determined for each of the three indices and correlations were calculated between mean seasonal flux and
each index.

**3    Results**

Identification of homogeneous snow cover days for the 2016 and 2017 snow seasons (June – October) resulted in
172 total days with 90 days occurring in the 2016 and 82 days in 2017. July, August, and September had the
highest number of classifiable days during the period. June and October still had periods with homogenous snow
cover, but they became intermittent and fewer classifiable days were in each of the months. This led to fewer
periods of study at the beginning and end of the snow seasons when snowpack was variable, with more in the late
winter and early spring months when snow cover was more consistent. Mean surface, cloud, and energy flux
characteristics of synoptic types identified during the two seasons are presented in Table 2.

**3.1    Synoptic types**

**3.1.1    Surface characteristics**

The dominance of the subtropical ridge in Australia's mid-latitudes is evident in the synoptic types. Four of the
types (T1,T2,T5 and T7) display dominant surface high pressure systems, each with slightly different orientation
and pressure centre locations (Figure 4) resulting in different energy flux characteristics. Dominant south-



southwesterly winds from T1 are the result of the high pressure centre being located to the northwest of the study
area. T2 has a predominantly zonal flow resulting from an elongated high to the north-northeast. T5 and T7 are
characterized by north-northwesterly flow from high pressure centres over the New South Wales
(NSW)/Queensland (QLD) coast and directly over the Snowy Mountains region, respectively.
T3 is characterized as having dominant northwest winds along a trough axis that is positioned over SEA with a
secondary coastal trough extending from southern NSW to the NSW/QLD border. T4 shows a transition from a
surface trough that has moved to the east of the study region to a high pressure system that is moving into the area
with winds from both features that converge over the Snowy Mountains region. The only synoptic type to have
dominant influence from a surface low was T6 that had weak south-southwesterly flow over the region from a
weak cut-off low to the east. For the purposes of this research, the identification of cut-off lows follows the
characteristics outlined by Chubb et al. (2011) that omits the presence of a closed circulation, but includes a cold
anomaly aloft that was cut off from the westerly wind belt.
Several types (T1, T4, T5, & T6) are considered to be 'transition types' that exist as the region is switching
between dominant high or low pressure surface features. T1, T4, and T6 are post-frontal transition types that show
high pressure ridging into the region following the passage of a trough that has either moved to the east (T1 &
T4) or developed into a weak lee-side cut-off low (T6). T5 shows the approach of a trough from the west and an
associated transition to a low pressure system. T2 and T7 show the area under the influence of zonal flow as a
result of high pressure systems centred over the area, while T3 shows SEA under the influence of a trough at the
time of observations.

### 3.1.2    Relative humidity and cloud cover

Understanding RH values associated with different synoptic types provides the ability to track types that are
favourable for high $Q_e$ exchange with the snowpack. In addition, RH values at all tropospheric levels can have
impacts on snowpack energy flux through influences on $K$ and $L$ exchange via changes to insolation and the
absorption and emission of $L$. The identification of RH characteristics and associated cloud cover is necessary to
fully develop energy flux characteristics for each type.
Many of the synoptic types display local RH maxima in the Snowy Mountains region at 850 hPa (Figure 5) and,
while T5 has the lowest RH values of all types, it still has slightly higher RH values over the area. The elevation
in RH values in the region is most likely caused by changes of airmass thermodynamic properties due to
orographic forcing of the mountains (Ahrens, 2012). T4 and T6 had the highest RH values over the region at 850
hPa with both being widespread and higher than 90%. T6 shows strong southerly advection of elevated RH values
from the tropics along the NSW and QLD coast ahead of troughs at 700 and 500 hPa (Figures 6 & 7) that are
associated with the surface cut-off low.
Manual identification of cloud cover agreed with the mean RH characteristics of T4 and T6 with both types having
100% cloud cover between partial and complete cloud cover days (Table 3). T6 showed the highest RH values of
any type with values greater than 90% over the region at the 700 and 500 hPa levels. While not definitive, this
would suggest that T6 has deeper or more cloud layers than T4, which likely only has clouds at lower altitudes.
T2 and T7 had the lowest percentage (both 76%) of days with any cloud cover, which is confirmed by their low
RH values at 700 hPa (<20% & <30%) and 500 hPa (<30% & <40%), respectively. In addition, they also had the



highest percentage of cloud-free days (both 23%). The remaining types (T1, T3, and T5) showed a relatively
consistent number of cloud days based on the satellite observations that were all above 85%.

### 3.1.3    Temperature

Temperature characteristics of synoptic types at low and mid-levels in the atmosphere are crucial to identify those
with the highest surface sensible heat flux characteristics. The highest mean temperatures and strongest warm air
advection (WAA) in the Snowy Mountains region at 850 hPa (Figure 8) was found to be from T5 that is driven
by converging winds on the back of a high pressure circulation to the east and the leading edge of a trough to the
west. T2 and T3 have the second and third highest temperatures, respectively, but have different advection
characteristics. T2 shows relatively weak WAA into the Snowy Mountains region associated with zonal flows at
850 hPa resulting from the high pressure circulations located to the north (similar to T7). However, T3 shows cold
air advection (CAA) from recent cold frontal passage with dominant winds from the west-northwest.
Overall, CAA at 850 hPa can be identified in four of the seven types (T1, T3, T4, and T6) and warm air advection
exists in the other three synoptic types (T2, T5, and T7). Of the four CAA types, T1 and T4 advection is being
generated through south-southwest and west-southwest winds, respectively, related to high pressure centres to the
northwest. Despite a stronger southerly component of dominant CAA winds in T1, temperatures are lower in T4
which has a higher westerly component to the wind. T6 shows CAA related to converging winds on the back of a
trough to the east and a high to the northwest.

### 3.1.4    Frequency and duration

The frequency of each synoptic type during the 2016 and 2017 snowpack seasons is shown in Table 4. T3 and T7
occurred most frequently with 26.74% (46 days) and 19.77% (34 days) respectively. The higher number of days
in T3 and T7 is reflected in the mean type duration that shows these types with the longest duration, which is
likely due to these synoptic types occurring in a more stagnant synoptic pattern over multiple days as seen in the
mean type duration (Table 2).
Transition probabilities for the 2016 and 2017 seasons were developed similar to those used by Kidson (2000)
that detail the likelihood of a synoptic type occurring on the following day given an initial type (Table 5). The
highest transition probabilities were identified for each type and a flowchart was developed based on the most
likely synoptic type progressions (Figure 9). If the highest transition probabilities were within < 0.05 of each
other, two paths were plotted. The flowchart shows what would be expected for a basic synoptic-scale circulation
at mid-latitudes; a trough propagating eastward into the Snowy Mountains region in T7, T5, and T3; either
continued eastward movement of the surface trough (T4) or the development of a weak cut-off low (T6); then
transitioning to dominant high pressure over the region again (T2, T1, or T7).

### 3.2    Energy flux characteristics of synoptic types

It is important to consider the effects of synoptic type frequency when determining primary sources of energy
fluxes over long periods as synoptic types that contribute the most to snowpack ablation may simply have a higher
rate of occurrence and lower daily energy flux values than other types. In order to obtain a more detailed
understanding of each type's energy flux, mean daily energy flux calculated for each type was determined to be a
better method of comparison. Therefore, both mean daily (Figure 10) and total snowpack period fluxes over the



two seasons (Figure 11) are discussed in MJ m⁻² to show synoptic type energy flux contributions made at short
and longer temporal scales.

### 3.2.1   Latent and sensible heat flux

Daily $Q_e$ was negative for each of the seven synoptic types and the magnitude of the values was shown to
correspond to the mean 850 hPa RH values for each type, which is due to an elevation of 1828 m A.S.L. at the
site that resulted in an ambient pressure of 791-831 hPa for the entirety of the study period. Two of the three types
with the lowest RH values (T2 and T5) showed the greatest negative latent heat flux and those with the higher RH
values (T1 and T6) showed the least amount of latent heat flux, which is consistent with conditions needed for
evaporation from the snowpack. T5 had the largest negative latent heat flux of any type with a mean daily value
of -1.26 MJ m⁻² day⁻¹ which corresponds to its low 850 hPa RH values, the highest observed surface mean daily
ambient temperature of 3.5 °C, and the second lowest observed surface mean RH value of 66% with only T2 being
lower (61%).
Overall, negative $Q_e$ was offset by positive $Q_h$ for most types with the exception of T3 that had mean surface
temperatures below zero (-1.01°C) and a measured surface RH value below 90% resulting in more latent heat loss
than sensible heat gain by the snowpack. Similar to trends seen in the $Q_e$, the highest daily mean $Q_h$ values were
associated with synoptic types with the highest temperatures at 850 hPa (T5, T7, & T2), which coincided with
observed temperatures from the energy flux tower (3.54°C, 1.42°C, & 1.04°C). T5 showed the highest daily $Q_e$
and $Q_h$ values as a result of having the highest temperatures and lowest RH values of any type. Ultimately, when
both turbulent terms are considered, T5 had the highest amount of energy flux into the snowpack (1.28 MJ m⁻²
day⁻¹) followed by T7 (1.11 MJ m⁻² day⁻¹) and T2 (0.97 MJ m⁻² day⁻¹).

### 3.2.2   Radiation flux

The largest contribution of radiative energy to the snowpack from all synoptic types was $K^*$ which accounted for
57-81% of total positive flux. By comparison, $L^*$ accounted for 66-90% of negative energy flux from the snowpack
with the highest amounts of loss belonging to the types with the lowest percentage of cloud cover (T1, T2, and
T3). Total radiation flux varied largely by synoptic type and was found to be positive in types T3, T6, & T5 and
negative for the rest of the types. The three types with positive net radiation had the highest incoming longwave
radiation flux values that allowed for greater cancellation of outgoing longwave values and allowed for incoming
shortwave radiation to dominate the net radiation flux. The largest loss in net radiation energy was exhibited by
T1 that was 3% higher than the next closest type (T2).  The types with net radiation loss (T1,T2, T4, and T7) had
values that ranged from -0.66 MJ m⁻² day⁻¹ (T4) to -1.44 MJ m⁻² day⁻¹ (T1). However, T4 had dissimilar cloud and
relative humidity characteristics to T2 and T7, which had the two lowest cloud cover percentages and two of the
lowest RH values. T4 had 100% cloud cover and had an associated reduction in incoming shortwave radiation
that allowed the outgoing longwave radiation term to become more dominant than in T2 or T7 and, therefore,
gave it the highest net radiative energy loss of the three.

### 3.2.3   Ground and precipitation heat flux

Energy flux from ground and precipitation sources were the smallest of any term for all types, with ground heat
and precipitation fluxes accounting for less than one percent of mean daily energy fluxes for all synoptic types.
Ground heat flux characteristics were similar between all synoptic types and varied little. While $Q_p$ was small



when examined as a daily mean value, it does show a high degree of variation that was associated with T5 and T3. This is due to several large rain events that occurred during 2016 (July 18; July 21-22; and August 31) and one during 2017 (August 15). Despite relatively low energy flux contributions by rainfall, it is interesting to note that the ten days with the highest rainfall fluxes (>0.05 MJ m$^{-2}$ day$^{-1}$) consisted of four T5 days, three T3 days, two T7 days, and one T6 day showing a significant clustering of high precipitation days in types T5 and T3.

### 3.2.4 Total daily net energy flux

Overall, four synoptic types (T3, T5, T6, and T7) had positive mean daily net energy flux to the snowpack (Figure 12). Of these, T5 had the largest energy flux that was related to its relatively high temperatures that contributed to the highest $Q_h$ value of any synoptic type and increased solar radiation from less cloud cover. Contrary to the reduction in cloud cover that aided T5 in having the highest total energy flux contributions, T6 had the highest cloud cover and yet had the second highest energy flux to the snowpack that was primarily due to increased incoming longwave radiation. Similarly, positive net radiation flux associated with T3 gave it a net positive daily net energy flux. Positive net energy flux from T7 is a result of relatively low percentage of cloud cover and the associated increase in $K\downarrow$ as well as the second highest $Q_h$ term of any type.

T1 and T4 showed the greatest negative mean daily net energy flux of all synoptic types (Figure 12), which could be attributed to their low $Q_h$ values as a result of the lowest measured temperatures of any synoptic type and to having low $K\downarrow$ terms. T2 also had a net negative mean daily energy flux but to a lesser extent than either T1 or T4. Relative humidity values lower than any other type were the primary driver behind T2's negative net value as it resulted in the highest longwave radiation loss from the snowpack through having the lowest cloud cover, as well as $Q_e$ loss.

T5 contributed the most energy to the snowpack during the two seasons despite T3 having nearly 96% more occurrences in the same period. This was largely due to high $Q_h$ values associated with strong WAA ahead of the passage of cold fronts associated with the T5 synoptic type, which had the second largest overall energy contribution to the snowpack. While mean daily energy flux contributions of T6 to snowpack energy flux are 103% higher than those of T3, the high number of occurrences associated with T3 made it the second highest contributor of energy flux during the two seasons with T6 contributing the third highest amount of energy flux. T7 had the smallest positive energy flux contribution to the snowpack during the two seasons. Types with negative seasonal fluxes showed similar magnitude to their mean daily values with T1 and T4 having the first and second highest negative flux, respectively, and T2 having the least negative flux.

All synoptic types had relatively large variation in mean daily net energy that can be attributed to the classification conducted by the k-means clustering technique. Each type consisted of classified days that had similar synoptic characteristics, but differences in system strength and position affected energy fluxes for individual days. Therefore, it is important to remember that each synoptic type is associated with a range of daily energy flux values in addition to the mean daily energy flux for each type.

### 3.2.5 Energy balance closure

Internal energy storage and melt processes can make measurement of energy balance closure over snow particularly difficult when internal measurements of the snowpack aren't available (Helgason and Pomeroy, 2012). As measurements of snowpack properties were not available during this period, assessment of energy





balance closure was conducted on periods when snow surface temperatures and ambient air temperature were less
than 0°C to limit the amount of energy being used in internal snowpack melt processes. Energy balance data was
divided into day and night periods following the methods of Helgason and Pomeroy (2012) that used an incoming
shortwave radiation threshold of $> 200$ Wm$^{-2}$ for day periods and $0$ Wm$^{-2}$ for night periods. Daytime periods
showed a slightly higher mean energy balance closure ratio of $0.33 \pm 0.55$ ($n = 761$) than night periods that had a
mean closure ratio of $0.31 \pm 0.51$ ($n = 2291$). Inability to account for nearly constant internal snowpack melt
processes has likely resulted in the low closure ratio and high variability in calculated energy balance closure.

### 3.3 Climatological trends of snowpack energy flux

#### 3.3.1 Frequency of occurrence and trends

Occurrence frequencies were determined for each synoptic type by classifying each day from June-October from
1979 to 2017. Statistics on seasonal averages of frequency for each type are displayed in Table 6 and show that
T2 and T3 have the greatest number of mean seasonal occurrences with 25 and 24 days per season, respectively.
All other types have between 20-23 days per season with the exception of T6 that has the lowest number of
seasonal occurrences at 17 days. T4 and T5 had the lowest variation between seasons of ±4 days indicating relative
consistency when compared to the other types that had variations of ±5-7 days per year.
Trends in occurrence frequencies of each type can be seen in Figure 13. In relation to the longer climatological
record, 2016 season showed a below average number of T1 days (characterised by high pressure) with only nine
days having the classification. In addition, it had the highest number of T3 days (characterised by pre-frontal
conditions) (tied with 2003) and was in the top six seasons for number of T6 events (cut-off low). The increase in
T3 events correspond with several rain-on-snow event days during the season that showed higher $Q_r$ contributions
than would be expected based on mean daily energy flux values. 2017 had synoptic type occurrences that were
close to climatological means with only T4 being slightly above the occurrence standard deviation with 28 total
days, which is above the normal mean seasonal occurrence of 21 ±4 days.
Significant variability exists between years, but linear trend analysis shows increases in frequency that can be
seen in T1, T2, T3, and T4 with decreases in T5, T6, and T7. T2 shows the greatest increase in frequency with a
gain of approximately six days annually over the 39 year period and T7 shows the largest decrease of five days
per season. All types associated with negative snowpack energy flux (T1, T2, and T4) have increases in their
seasonal frequency and T3 was the only type associated with positive snowpack energy flux that showed an
increase in frequency. The types associated with the largest quantities of positive snowpack energy flux (T5 and
T6) showed declines of 1-2 days per season.
Snowpack energy flux pertaining to the June-October seasons (1979-2017) was estimated using the identified
synoptic types and mean daily energy flux characteristics of each type identified from the 2016 and 2017
observational period. Similar to trends seen in the frequency of synoptic types that generally shows types
associated with positive (negative) snowpack energy flux decreasing (increasing), overall estimated mean
seasonal snowpack energy flux is shown to decrease by 7.41 MJ m$^{-2}$ over the 39 year period (Figure 14). However,
estimated mean seasonal snowpack energy flux was still strongly positive for all years with only 1997 and 2013
having positive fluxes less than 10 MJ m$^{-2}$, which also showed the highest numbers of T1 and T2 that are
characterized as having dominant high pressure over the region.





### 3.3.2 Atmospheric teleconnections

Investigation into teleconnection impacts on estimated seasonal snowpack energy flux showed relationships with IOD and SOI phase, but little relationship with SAM phase (Figure 15). A negative (positive) correlation with an $R^2$ value of 0.47 (0.38) can be seen between IOD (SOI) phase and total seasonal snowpack energy flux, which agrees with the findings of Pepler et al. (2014) who showed IOD as a greater influence on cool season precipitation in SEA than ENSO. Of the ten seasons with the largest estimated positive snowpack energy flux, 7 occurred during a negative IOD phase and 9 occurred during a positive SOI phase. Six of these seasons had abnormally high T6 (cut-off low) counts that all fell outside of one standard deviation from the mean of $17 \pm 6$ days and seven seasons had low T1 (high pressure). This indicates that the highest estimated snowpack energy flux seasons tend to occur during a negative IOD phase and positive SOI phase through a shift from high pressure occurrences to dominant cut-off lows over the region.

Seasons with the highest estimated positive snowpack energy flux were distributed over the climatological study period with one occurring during 1979, four in the 1980s, one in the 1990s, three in the 2000s, and one in the 2010s. Periods of reduced energy flux are seen in the 1990s and 2010s (Figure 14), which can be attributed to dominance of positive IOD phase and negative SOI phase during these periods.

Overall, IOD phase was the dominant climatological control of estimated seasonal energy flux and accounted for an average increase of 31% when changing from a positive to negative phase (Table 7). SOI phase had a similar effect accounting for a mean positive energy flux increase of 28% during seasons that were positive. While SAM phase can account for an increase of 12% energy flux by season when it is negative, its contribution is relatively small compared to the other two terms. Negative IOD and positive SOI can account for a 48% increase in energy flux to the snowpack when compared to years with opposite phases. The highest estimated energy flux to the snowpack occurs when IOD is negative, SAM is negative, and SOI is positive, which results in up to 56% more energy flux to the snowpack than other periods. While the highest difference in energy flux is seen in the correct combination of IOD -, SAM -, and SOI + periods when compared to others, only 40% of the top ten seasons with the largest energy flux had this characteristic. However, 80% (including three of the top five seasons) had a negative IOD and positive SOI suggesting that strength of phase is also an important consideration.

## 4 Discussion

### 4.1 Properties of synoptic type energy balance

Net shortwave radiation flux contributed the largest amount of energy to the snowpack for all synoptic types ranging from 57-81% of mean daily energy flux with T5 being the only synoptic type below 60% contribution (57%) of $K^*$ to the snowpack. These results agree with Fayad et al. (2017) who noted that radiative fluxes are the dominant source of snowpack melt energy in mountain ranges with Mediterranean climates. Net sensible heat flux contributed the second highest percentage of mean daily energy flux to the snowpack accounting for 13-40% of positive fluxes. The largest contributions of $Q_h$ to the snowpack are associated with synoptic types T2, T5, and T7 that are characterised by high pressure and northwesterly winds that are associated with WAA. Hay and Fitzharris (1988) noted that, while radiative terms were responsible for the majority of energy contributions to glacier melt in New Zealand's Southern Alps, turbulent fluxes contributed significant amounts of energy to melt. Similarly, despite $Q_h$ not being the dominant energy flux to the snowpack for any synoptic type, it does account



for nearly half of the energy flux to the snowpack for T5 (40%) and over a third for T2 (34%), and is still a
significant source of energy flux to the snowpack for all synoptic types.
Mean daily energy loss from the snowpack was from latent heat and net longwave radiation, which dominated
T1, T2, and T4 resulting in negative mean daily energy fluxes from the snowpack. Net longwave radiation was
the most influential term in the emission of energy from the snowpack accounting for 66-90% of energy loss with
net latent heat flux accounting for 10-27% of outgoing energy flux. Though the methodology of this paper
distinguishes between shortwave and longwave fluxes in order to better examine the effects of synoptic-scale
features such as RH or cloud cover on radiative transfers, many prior works have not made the same distinction
in terms (Hay and Fitzharris, 1988; Moore and Owens, 1984; Neale and Fitzharris, 1997; Stoy et al., 2018). It
should be noted that had $Q^*$ been used for comparison, the results of this paper agree with several studies (Bednorz,
2008b; Moore and Owens, 1984; Sade et al., 2014) that found that turbulent fluxes were the dominant fluxes when
examining the energy flux characteristics on snowpacks in climates similar to that of the Snowy Mountains.
Mean daily $Q_g$ values were found to account for only a small fraction of total energy flux to the snowpack
consisting of 1-4% of daily positive energy fluxes. Similarly, energy flux to the snowpack from rainfall has been
shown to only contribute ~1% of total seasonal energy flux for five of the seven synoptic types which agrees with
the findings of other studies (Bilish et al., 2018; Mazurkiewicz et al., 2008). However, precipitation was
responsible for ~2% of the daily mean energy flux of the two synoptic types primarily associated with rain-on-
snow events, T5 and T3 respectively. Although fluxes imparted on the snowpack from rainfall are relatively small
when compared to all positive fluxes, the accompanying energy flux characteristics of the synoptic types
associated with rain-on-snow events are responsible for two of the three largest contributions of overall snowpack
energy fluxes.

## 4.2 Synoptic patterns and energy flux

Snowpack energy flux characteristics recorded at the Pipers Creek catchment headwaters have been related to
synoptic weather types that occurred during the 2016 and 2017 snow seasons. The resulting analysis reveals a
maximum in positive energy flux as pre-frontal troughs approach the Snowy Mountains, followed by cold front
conditions during the T7→T5→T3 common progression pattern identified here. Several factors cause high
positive energy flux during these periods that include: an increase in temperatures due to WAA and the associated
increase in positive $Q_h$; decrease in negative $L^*$ due to an increase in cloud cover; a decrease in $Q_e$ following
frontal passage and associated increase in RH; and progressively increasing $Q_r$ as the trough approaches and
immediately after passage.
Synoptic types characterized by surface high pressure as their primary influence (T1, T2, T4, and T7) showed the
lowest contributions to snowpack energy flux. In T1, T2, and T7, net shortwave radiation terms ($K^*$) were positive
and varied by ~2-24% for these types, however, low RH and cloud cover allowed for highly negative $L^*$ terms
that were not compensated by change in $K^*$. In contrast, T4 had higher cloud cover and increased RH that were
due to advection of moisture from the Tasman Sea. The higher RH in T4 and low mean air temperature (-2.06°C)
resulted in $Q_e$ and $Q_h$ terms of similar magnitudes, but opposite signs that nearly cancelled out. This resulted in a
$L^*$ term that was of lesser magnitude than those of T1, T2, and T7, but still the dominant term in its energy
exchange.



Four primary synoptic circulation patterns were identified during the study period. Each of the four patterns and their associated energy flux values calculated from mean daily flux and mean type duration can be seen in Figures 9 and 16. While each pattern differs towards the end of the cycle, each one has the T7→T5→T3 progression in common. Unsurprisingly, the highest contribution of mean energy flux to the snowpack (4.95 MJ m$^{-2}$) is from Pattern 1, which lacks the synoptic types associated with negative snowpack energy flux that the other three patterns contain. Pattern 3 had the lowest contribution to snowpack energy flux (1.86 MJ m$^{-2}$) due to it containing types with the highest net energy loss (T1 and T4), but the positive flux types (T3, T5, and T7) supplied enough energy to outweigh the negative flux of T1, T2, and T4 and make the overall pattern energy flux positive.

### 4.3 Climatological trends in estimated energy flux

Estimated positive energy flux decreases over the last 39 years result from reductions in T5, T6, and T7. These types accounted for a significant portion (29%) of rain and snow during the 2016-2017 study period with only T3 contributing more precipitation (59%). These types all share a common characteristic of an intense surface high-amplitude trough or weak cut-off low. In contrast, increases are seen in T1 and T2 that are characterized by negative snowpack energy flux suggesting a shift from troughs to high pressure zonal characteristics that has also been noted in other parts of Australia (Hope, 2006).

Reductions in winter precipitation-generating synoptic types agree with other literature (Chubb et al., 2011; Fiddes et al., 2015; Theobald et al., 2016) that showed a reduction in the number of days with precipitation. As T5, T6, and T7 are three of the four synoptic types with positive energy flux to the snowpack, the reductions in precipitation and associated reductions in energy flux available for snow melt for these types may result in a net-zero change in the energy contributed to existing snowpack. This indicates a possibility that reductions in snowpack depth and duration (Hennessy et al., 2008) are primarily a result of reductions in precipitation rather than increases in energy fluxes to the snowpack.

Phases of the IOD and SOI showed significant impacts on energy flux to the snowpack through changes to the frequency of synoptic types, but SAM was found to have relatively little influence similar to findings by Meneghini et al. (2007). IOD was found to be the more strongly correlated with energy flux patterns which is aligned with previous research (Pepler et al., 2014; Ummenhofer et al., 2009) that showed IOD as the dominant control of SEA rainfall variability when compared to ENSO. Increases in energy flux to the snowpack were primarily due to reductions of T1 and its associated negative energy flux and increases of T6 that has the second highest positive energy flux of all synoptic types and is associated with precipitation. Despite potential increases in precipitation in SEA during periods of negative IOD (Ashok et al., 2003), energy flux to the snowpack reaches peak values resulting in faster melt of the snowpack that accumulates.

### 5 Conclusions

Overall, periods of pre-cold frontal passage contribute the most energy fluxes to snowpack melt due to warm air advection ahead of the front, a reduction in cloud cover allowing for higher incoming shortwave radiation, and the gradual development of precipitation that often contributes to rain-on-snow events. Estimated snowpack energy flux in the Snowy Mountains shows a decreasing trend over the past 39 years, but reductions in snowpack duration (Hennessy et al., 2008) are still seen due to reductions in cool-season precipitation (Chubb et al., 2011; Theobald et al., 2016). The influence of IOD, SOI, and SAM (to a lesser extent) phase to snowpack energy flux generally manifests as a shift from anti-cyclonic characteristics and lower cloud cover values seen in T1 to

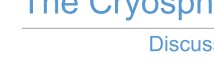
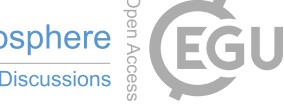

cyclonic conditions associated with the cut-off low of T6 and 100% cloud cover. The identification and
understanding of these climate patterns and their impact on synoptic systems is an important step towards
determining snowpack energy flux and should be closely monitored as changes can indicate significant increases
in snowpack ablation.
Reductions to precipitation and snowpack energy flux can be expected to continue into the future due to impacts
from anthropogenic climate change on atmospheric circulation from a poleward shift of the sub-tropical jet
(Kushner et al., 2001) and winter-time shifts in the position and intensity of the sub-tropical ridge (Larsen and
Nicholls, 2009). These changes will likely result in further reduction in precipitation in the Snowy Mountains and
reductions in snowpack depth and duration. Australia's modelled reduction in runoff and snowmelt as a result of
climate change progression (Adam et al., 2009) reinforces the importance of understanding the Australian
snowpack and its causes of ablation.
While this work was conducted solely on the Australian snowpack, snowpacks in other regions such as New
Zealand (Hay and Fitzharris, 1988; Neale and Fitzharris, 1997), Canada (Romolo et al., 2006a; 2006b), and the
Arctic (Drobot and Anderson, 2001) see similar synoptic-scale effects on snowmelt to those presented here. An
increased burden on freshwater systems for agriculture, drinking water, and energy production will continue as
anthropogenic climate change and its associated effects progress (Parry et al., 2007). Therefore, continued work
on marginal snowpack ablation processes, such as those within the forested regions of Australia's Snowy
Mountains, will be important to resource management and should be explored.
**Data Availability**
All energy flux data and code used for this project (with the exception of precipitation data) can be obtained by
contacting Andrew Schwartz (Andrew.Schwartz@uq.edu.au). ERA-Interim reanalysis data used in this study are
freely available from the European Centre for Medium-Range Weather Forecasts
(https://www.ecmwf.int/en/forecasts/datasets/reanalysis-datasets/era-interim).
**Author Contributions**
AS, HM, AT, and NC designed the experiments and AS conducted them. AT developed the k-means clustering
and synoptic typing code. AS developed the code related to energy balance and eddy covariance measurements,
trends, and teleconnections. AS prepared the manuscript.
**Competing Interests**
The authors declare that they have no competing interests.
**Acknowledgements**
The authors would like to thank Shane Bilish for establishment of the Pipers Creek snowpack research catchment,
Michael Gray for installation and maintenance of the energy balance tower, and the Weather and Water team at
Snowy Hydro Limited for their contributions of data and field support during the data collection and analysis
process.





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













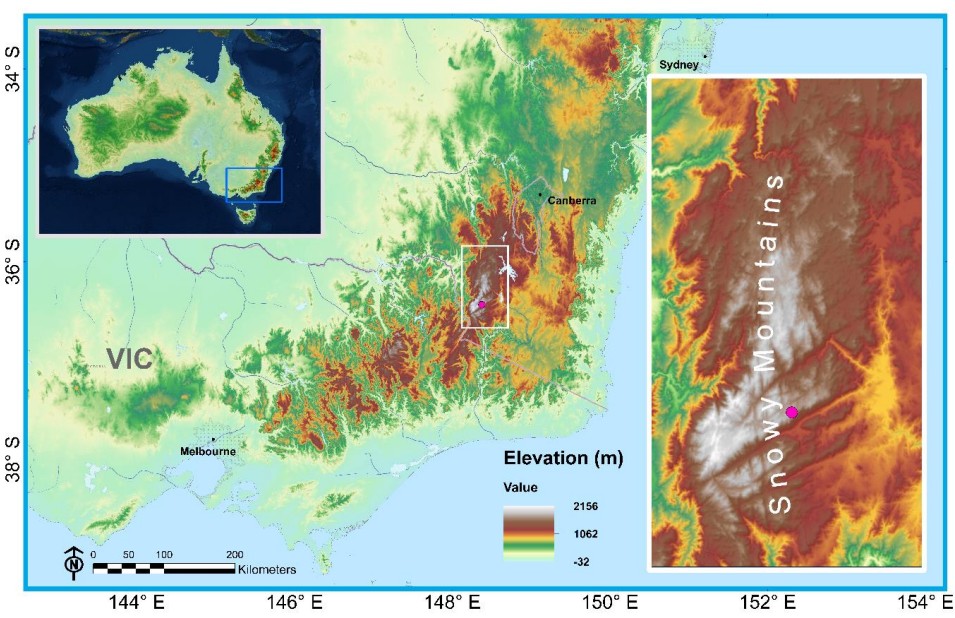

**Figure 1: Map of southeast Australia and the Snowy Mountains. Pink dot represents the location of the energy balance instrumentation site.**













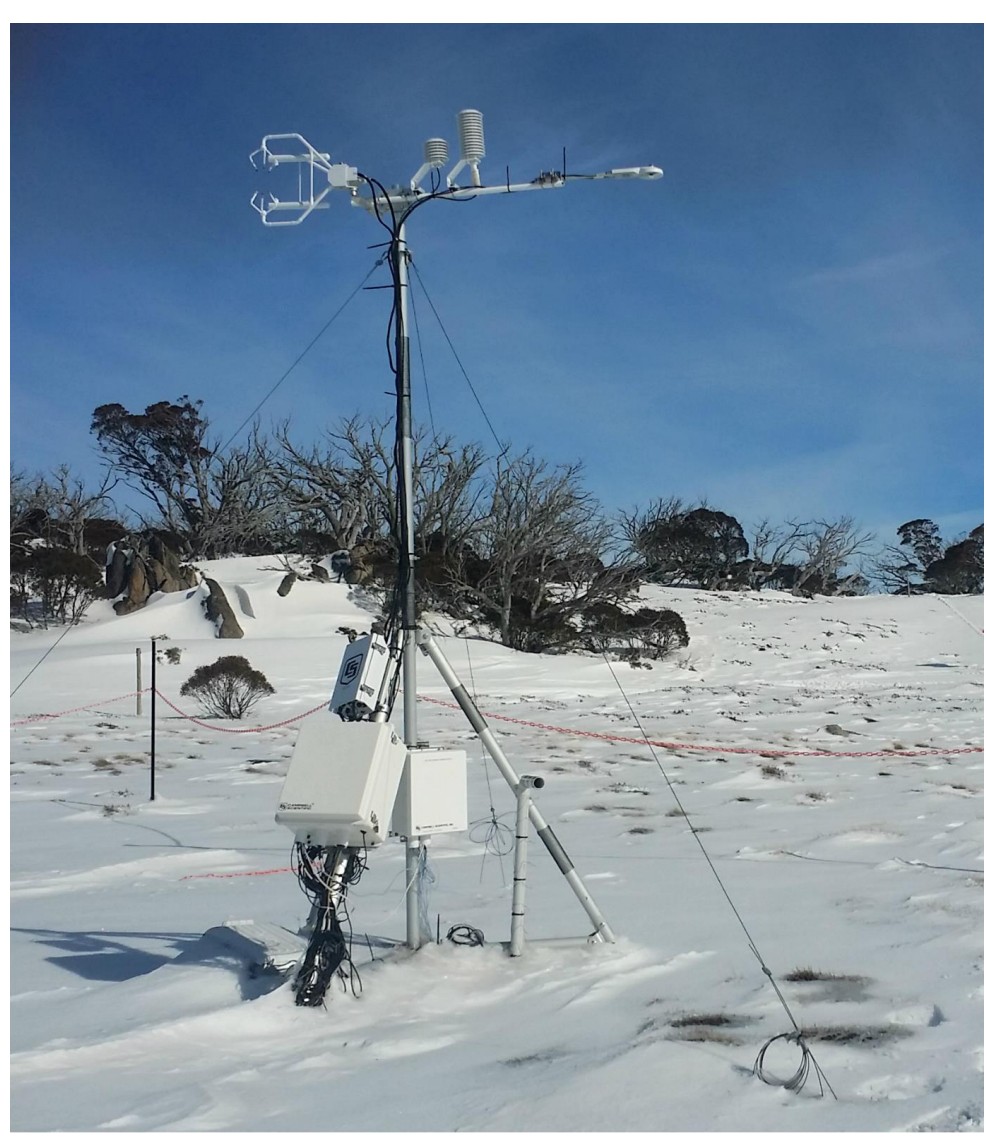

**Figure 2: Energy balance field site with eddy covariance instrumentation at Pipers Creek catchment headwaters.**














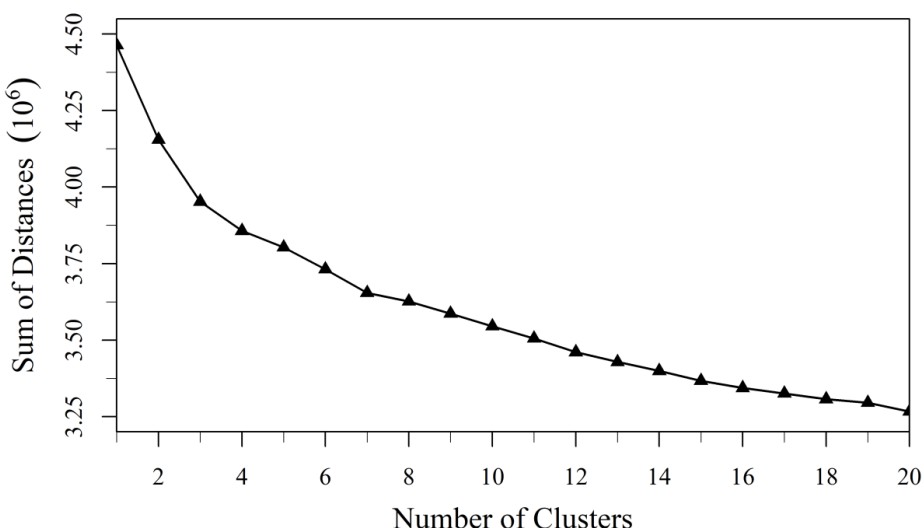

**Figure 3: Plot of cluster distances from centroid with a clear elbow at 7 clusters illustrating the optimum number to be**
**used in analysis.**


















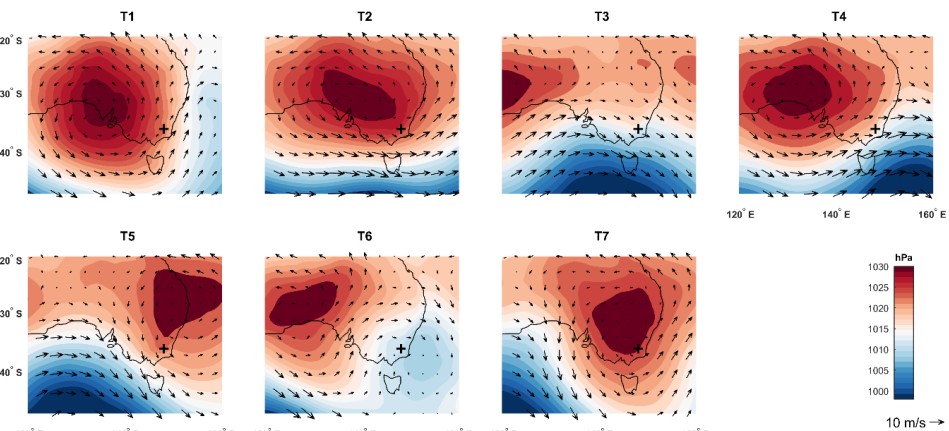

**Figure 4: Mean MSLP and 10m wind vectors for each synoptic type over the southeast Australia region for the 2016**
**and 2017 seasons. Location of surface energy balance site marked with '+'.**






















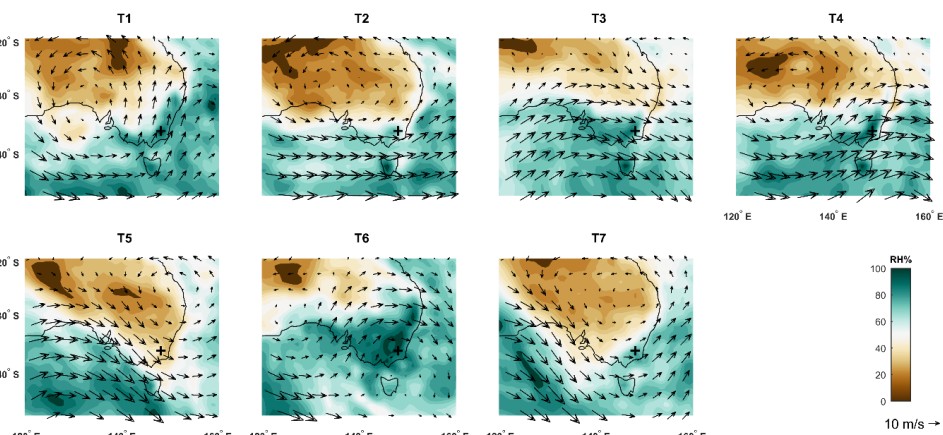

**Figure 5: Mean 850 hPa relative humidity values and wind vectors for each synoptic type over the southeast Australia**
**region for the 2016 and 2017 seasons. Location of surface energy balance site marked with '+'.**






















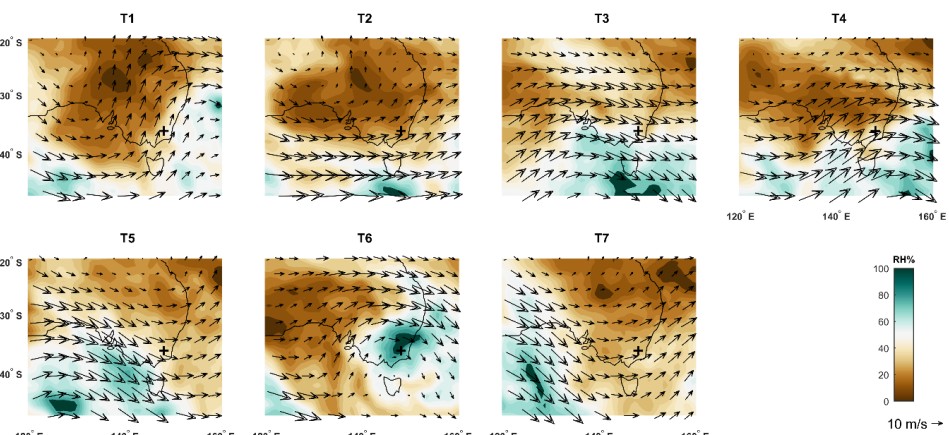

**Figure 6: Mean 700 hPa relative humidity values and wind vectors for each synoptic type over the southeast Australia**
**region for the 2016 and 2017 seasons. Location of surface energy balance site marked with '+'.**






















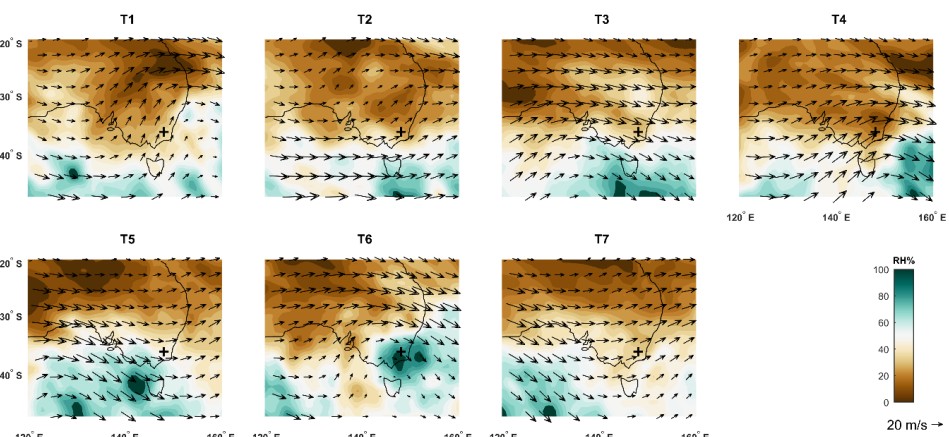

**Figure 7: Mean 500 hPa relative humidity values and wind vectors for each synoptic type over the southeast Australia**
**region for the 2016 and 2017 seasons. Location of surface energy balance site marked with '+'.**






















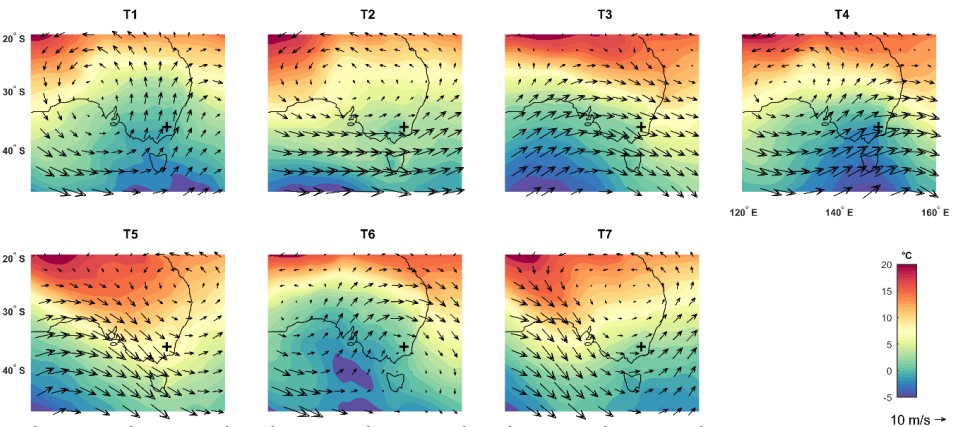

**Figure 8: Mean 850hPa temperature values and wind vectors for each synoptic type over the southeast Australia region**
**for the 2016 and 2017 seasons. Location of surface energy balance site marked with '+'.**





































**Figure 9: Flowchart of four primary synoptic type patterns/progressions based on probability of transition for the 2016**
**and 2017 seasons.**











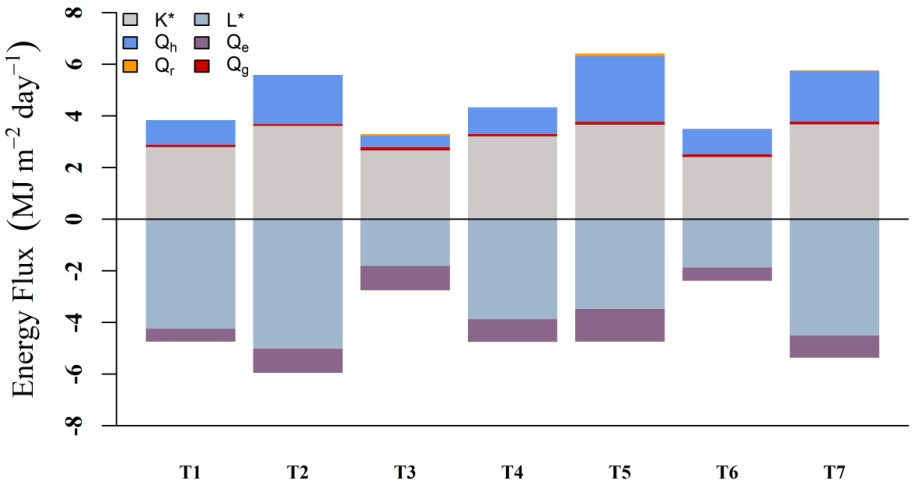

**Figure 10: Mean daily snowpack energy fluxes by term for each synoptic type for the 2016 and 2017 seasons.**
















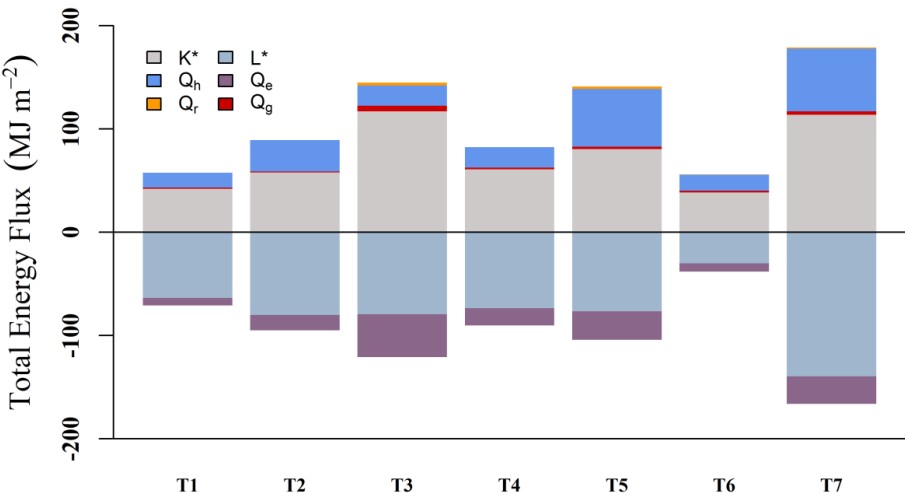

**Figure 11: Total snowpack energy fluxes by term for each synoptic type during study period for the 2016 and 2017 seasons.**



















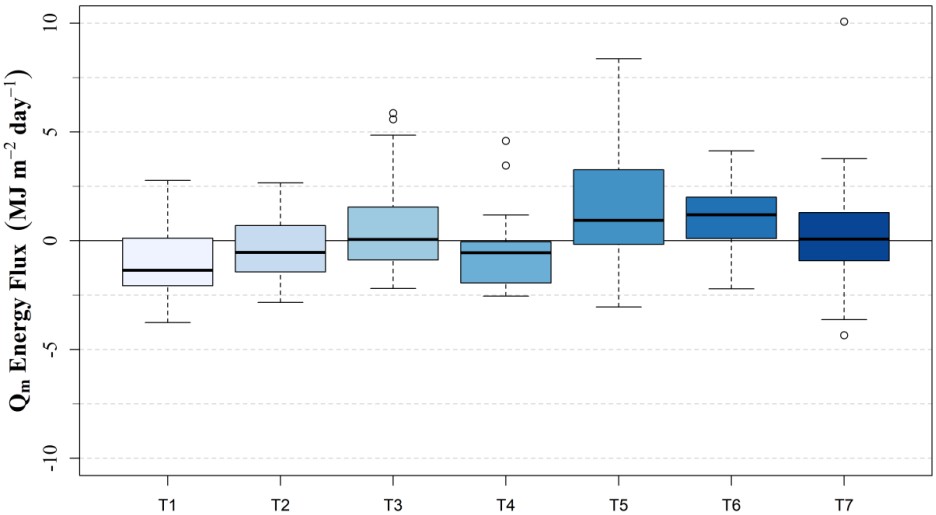

**Figure 12: Box and whisker plot of daily snowpack energy fluxes by synoptic type for the 2016 and 2017 seasons.**






















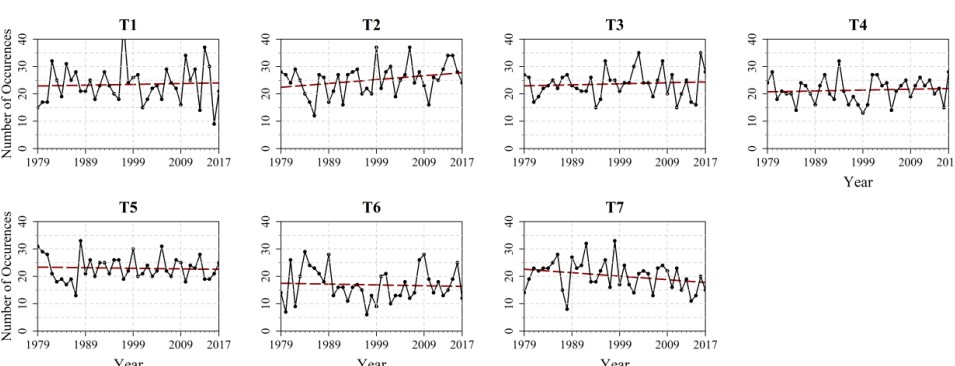

**Figure 13: Climatological trends in the seasonal frequency of each synoptic type from 1979 to 2017 indicated by the**
**red dashed line.**


















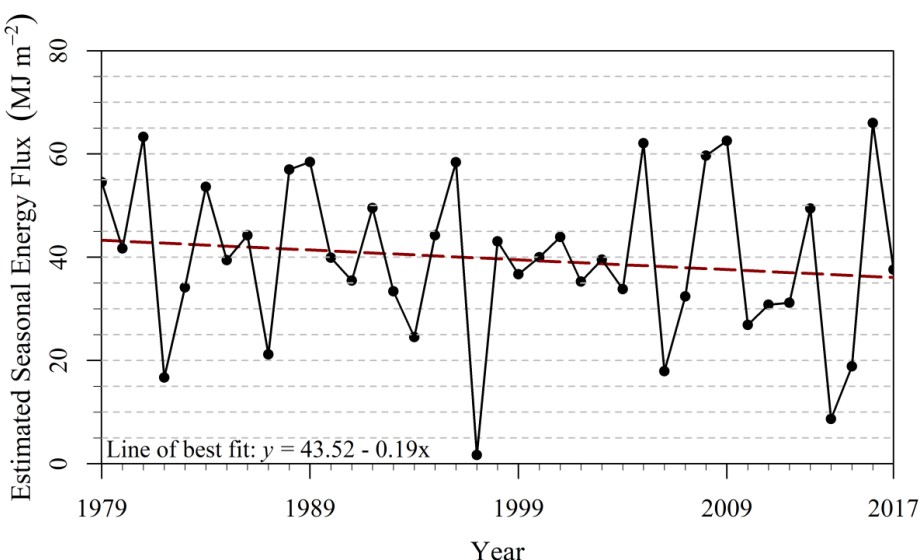

**Figure 14: Trend in estimated seasonal snowpack energy flux.**




































**Figure 15: Correlation between Seasonal Snowpack Energy Flux and Indian Ocean Dipole (IOD) phase (a), Southern**
**Annular Mode (SAM) phase (b), and Southern Oscillation Index (SOI) phase (c).**





**Figure 16: Calculated synoptic pattern snowpack fluxes based on mean daily values and mean duration of synoptic type.**




| Instrument | Manufacturer | Variables Measured | Accuracy |
|---|---|---|---|
| SI-111 | Apogee Instruments | Surface Temperature ($T_{sfc}$) | ± 0.2°C -10°C<T<65°C ± 0.5°C -40°C<T<70°C |
| CS650 | Campbell Scientific | Soil Water Content (SWC) | ± 3% SWC |
| | | Soil Temperature | ± 5°C |
| CSAT3A | Campbell Scientific | Wind Components ($u_x$, $u_y$, $u_z$); Wind Speed (u) and Direction (°); and Sonic Temperature | ± 5 cm s$^{-1}$ |
| EC150 | Campbell Scientific | $H_2O$ Gas Density | 2% |
| NOAH II | ETI Instrument Systems | Precipitation Accumulation | ± 0.254 mm |
| HFP01 | Hukseflux | Soil Heat Flux | < 3% |
| CNR4 | Kipp and Zonen | K↓, K↑, L↓, L↑ | K < 5% Daily Total L < 10% Daily Total |
| HMP155 | Vaisala | Air Temperature ($T_d$) | < 0.3°C |
| | | Relative Humidity (RH) | <1.8% RH |
| PTB110 | Vaisala | Barometric Pressure | ± 0.15 kPa |


**Table 1: Information on instruments used at the Pipers Creek catchment site.**


















| Synoptic Type | T1 | T2 | T3 | T4 | T5 | T6 | T7 |
|---|---|---|---|---|---|---|---|
| Surface Characteristics | High pressure; SW winds | High pressure; WNW winds | Frontal; NW winds | High/low transition; W winds | High Pressure; NNW winds | Lee-side low; SW winds | High pressure; WNW winds |
| Cloud Cover (% days with any cover) | 87.50% | 76.47% | 89.13% | 100.00% | 87.50% | 100.00% | 76.47% |
| $Q_h$ (MJ m$^{-2}$ day$^{-1}$) | 0.96 | 1.91 | 0.44 | 1.04 | 2.54 | 0.98 | 1.97 |
| $Q_e$ (MJ m$^{-2}$ day$^{-1}$) | -0.49 | -0.94 | -0.94 | -0.88 | -1.26 | -0.51 | -0.86 |
| K↓ (MJ m$^{-2}$ day$^{-1}$) | 12.57 | 15.17 | 9.39 | 13.47 | 12.94 | 9.33 | 12.93 |
| K↑ (MJ m$^{-2}$ day$^{-1}$) | -9.77 | -11.55 | -6.72 | -10.26 | -9.28 | -6.91 | -9.25 |
| L↓ (MJ m$^{-2}$ day$^{-1}$) | 20.08 | 20.14 | 24.78 | 21.89 | 23.66 | 24.66 | 21.80 |
| L↑ (MJ m$^{-2}$ day$^{-1}$) | -24.32 | -25.15 | -26.59 | -25.76 | -27.14 | -26.53 | -26.31 |
| $Q_g$ (MJ m$^{-2}$ day$^{-1}$) | 0.08 | 0.06 | 0.13 | 0.09 | 0.11 | 0.10 | 0.10 |
| $Q_r$ (MJ m$^{-2}$ day$^{-1}$) | 0.00 | 0.00 | 0.06 | 0.00 | 0.10 | 0.01 | 0.02 |
| $Q_m$ (MJ m$^{-2}$ day$^{-1}$) | -0.89 | -0.36 | 0.55 | -0.42 | 1.68 | 1.12 | 0.40 |
| $Q_m$ Standard Deviation (MJ m$^{-2}$ day$^{-1}$) | 1.96 | 1.53 | 1.91 | 1.89 | 2.72 | 1.81 | 2.57 |
| Mean Type Duration (Days) | 1.23 | 1.31 | 1.59 | 1.19 | 1.20 | 1.33 | 1.42 |


**Table 2: Mean daily characteristics and values for each synoptic type.**














| | Number of Days | Days with No Cloud Cover | Days with Partial Cloud Cover | Days with Complete Cloud Cover | Days with Any Cloud Cover |
|---|---|---|---|---|---|
| Type 1 | 16 | 12.50% | 43.75% | 43.75% | 87.50% |
| Type 2 | 17 | 23.53% | 52.94% | 23.53% | 76.47% |
| Type 3* | 46 | 8.70% | 23.91% | 65.22% | 89.13% |
| Type 4 | 19 | 0.00% | 42.11% | 57.89% | 100.00% |
| Type 5 | 24 | 12.50% | 58.33% | 29.17% | 87.50% |
| Type 6 | 16 | 0.00% | 6.25% | 93.75% | 100.00% |
| Type 7 | 34 | 23.53% | 32.35% | 44.12% | 76.47% |

*One day was omitted from classification due to lack of data


**Table 3: Cloud cover characteristics of synoptic types.**






















|  | Number of Days | Percentage of Events |
|---|---|---|
| Type 1 | 16 | 9.30% |
| Type 2 | 17 | 9.88% |
| Type 3 | 46 | 26.74% |
| Type 4 | 19 | 11.05% |
| Type 5 | 24 | 13.95% |
| Type 6 | 16 | 9.30% |
| Type 7 | 34 | 19.77% |


**Table 4: Frequency and percentages of synoptic types.**






















|  | To | | | | | | |
|---|---|---|---|---|---|---|---|
| From | T1 | T2 | T3 | T4 | T5 | T6 | T7 |
| T1 | 0.28 | 0.22 | 0.00 | 0.00 | 0.06 | 0.00 | 0.44 |
| T2 | 0.12 | 0.24 | 0.12 | 0.12 | 0.06 | 0.06 | 0.29 |
| T3 | 0.07 | 0.02 | 0.39 | 0.26 | 0.00 | 0.22 | 0.04 |
| T4 | 0.24 | 0.29 | 0.12 | 0.12 | 0.12 | 0.00 | 0.12 |
| T5 | 0.00 | 0.08 | 0.56 | 0.04 | 0.20 | 0.00 | 0.12 |
| T6 | 0.27 | 0.07 | 0.07 | 0.00 | 0.13 | 0.20 | 0.27 |
| T7 | 0.00 | 0.24 | 0.24 | 0.03 | 0.42 | 0.03 | 0.27 |


**Table 5: Probability of transition from one synoptic type to another on the following day. Note that T3's most likely**
**transition is to itself (consecutive days of the same synoptic type).**




















|        | Mean | Standard Deviation | Median |
|--------|------|--------------------|--------|
| Type 1 | 23   | 7                  | 23     |
| Type 2 | 25   | 6                  | 26     |
| Type 3 | 24   | 5                  | 24     |
| Type 4 | 21   | 4                  | 21     |
| Type 5 | 23   | 4                  | 22     |
| Type 6 | 17   | 6                  | 16     |
| Type 7 | 20   | 5                  | 21     |


**Table 6: Statistics on seasonal frequency of synoptic types from 1979 through 2017.**
























| | Phase | | Estimated Mean Seasonal Flux (MJ m$^{-2}$) | No. of Seasons |
|---|---|---|---|---|
| IOD + | SAM + | - | 33.20 | 16 |
| IOD + | SAM - | - | 34.99 | 11 |
| IOD - | SAM - | - | 55.35 | 6 |
| IOD - | SAM + | - | 50.11 | 6 |
| | | | | |
| IOD + | SOI + | - | 40.91 | 12 |
| IOD + | SOI - | - | 28.34 | 15 |
| IOD - | SOI - | - | 45.63 | 2 |
| IOD - | SOI + | - | 54.15 | 10 |
| | | | | |
| SAM + | SOI + | - | 45.74 | 13 |
| SAM + | SOI - | - | 26.36 | 9 |
| SAM - | SOI + | - | 48.64 | 9 |
| SAM - | SOI - | - | 34.90 | 8 |
| | | | | |
| IOD + | SAM + | SOI + | 41.99 | 7 |
| IOD + | SAM + | SOI - | 26.36 | 9 |
| IOD + | SAM - | SOI + | 39.39 | 5 |
| IOD + | SAM - | SOI - | 31.32 | 6 |
| IOD - | SAM + | SOI + | 50.11 | 6 |
| IOD - | SAM + | SOI - | 0.00 | 0 |
| IOD - | SAM - | SOI + | 60.21 | 4 |
| IOD - | SAM - | SOI - | 45.63 | 2 |


**Table 7: Mean seasonal snowpack energy flux values associated with various IOD, SAM, and SOI phase combinations.**





