# Peer review of "Quantifying the impact of synoptic weather types, patterns, and trends on energy fluxes of a marginal snowpack 2"

_The Cryosphere, 2019_

## Referee Comment (RC1) · Mathias Rotach (Referee) · 2 May 2019

General considerations

In this paper, the authors set out to investigate the impact of the synoptic flow conditions (and their changes over the years) to available energy for a snow pack in South East Australia. This is, first of all, a very valuable undertaking and adds to providing additional scientific understanding on potential causes and mechanisms of impact (going beyond the simplistic global change → warmer → more snow melt). For this purpose, the authors use surface data from one energy balance station (situated on the snow pack) in connection with ERA-Interim reanalysis data for the synoptic situation. It

seems to me that in both these data sources, there are conceptual problems that need to be addressed before the paper can be recommended for publication. I usually, when preparing my review (i.e., reading the paper) list 'major' and 'minor' issues separately (this is what can be found below). Still, I add those two critical issues separately – even if some of the 'major' (and even some of the minor) comments address the same topics.

I have tried to give some references for specific points raised – and they often happen to be from my own work. This is not 'to make the authors cite my papers' – it is just that it is the quickest way to get this put together. Often, there are other, equally suited papers around for the same issue.

Critical issues

1) The authors use one surface EB station in an area of large spatial inhomogeneity, and derive all their results from this one site (2 years ['seasons'] of measurements). The individual contributions to the EB are then attributed to each synoptic type and further used in a climatological study. All this, basically prompts the questions: i) how representative is this one site for any larger area? Is the very local measurement representing the [energy distribution] in the 'Snowy Mountains' to any degree? ii) Are the energy estimates accurate enough to draw any firm conclusions? As for ii) the authors take most of the necessary corrections etc. to measurements appropriately into account and also apply a quality assurance procedure. However, this leads to the necessity to basically 'produce' 50% of the data (and I am not convinced that a lookup table is the best solution to do this, if necessary, see below) with a claimed uncertainty of some 21 W/m2 (latent heat flux) and 34 W/m2 (sensible heat flux). Summed over the day this amounts to some 1.8 (2.9) MJ/m2 (and day). Looking at Fig. 10 reveals that this is more for both Q_e and Q_H than the difference between the synoptic pattern with the largest and the smallest contribution to the EB. So, this triggers two consequences: First, the accuracy of the measurements has to be increased (which means not to use the Campbell standard post processing, taking into account (estimating)

the uncertainty due to assumptions which are not particularly good (such as 'constant fluxes', such as planar fit coordinate rotation with only one plane', etc. – see some of the 'major comments'). Second, the estimation of missing data has to become better (I am pretty sure that much better accuracy can be obtained by using a sophisticated statistical model), and finally, the uncertainty of the estimates has to be taken into account when discussing the results (e.g. Fig. 10). For example, this 'bar plot' could include error bars.

Concerning the representativeness question (i), I can only say, that our own measurements of contributions to the EB in complex terrain (e.g. Rotach et al. 2017), yields huge differences in all the contributions (with the exception of the longwave balance) from site to site, at kilometre scale. We haven't investigated the impact on daily means (or sums), but looking at daily cycles shows that they can be substantial (several 100 W/m2 in the maximum and several hours of difference when the maximum occurs, i.e. when the fluxes start to 'go back'). I have no useful suggestion, how to actually estimate the potential attribution error that might be introduced when using only one site (based on published results, that is) – but at least, I think it must be discussed (maybe Bellaire et al. 2017 provide some hints on orders of magnitude).

2) My second concern is the treatment of 'days' as an entity. Each day is identified with a 'synoptic pattern' (which is determined based on mean daily values of SLP, temperature, humidity, wind, etc.). While I can see how statistically this procedure yields a synoptic pattern for each day, I do not understand, why those patterns should always start at midnight and last for a day. Looking at the patterns shows that they constitute 'snapshots' of a dynamic development (e.g. the T7-T5-T3 sequence (Fig. 9) actually corresponding to an eastward moving trough to the south of the area of interest). This means that those patterns are only 'statistical entities' with the actual position of the trough on a given day (00.00 to 24.00 o'clock) being 'closest' to one of the patterns. If the trough had built (or moved in) at 12.00 o'clock (rather than at midnight and 'stayed there' for 24 hrs) as manifested in T7, the actual pressure

distribution (on day two, say) would likely be somewhere between T7 and T5 – with possibly quite drastic consequences for the local pressure (geopotential) distribution (e.g. Fig. 4) and hence advection pattern. This, in turn, will (potentially) affect large parts of the explanations (e.g., the WAA and CAA as discussed in Section 3.1.3). Can the authors 'defend' their choice of 24 hr time segments in some more detail and comment on the potential consequences (day-to-variability within a given pattern)?

Major comments

1) L. 131: It is good to provide the information on the instrumentation. It is also necessary, however, to provide information on data post-processing. Later (l. 217 ff), I see most of it is mentioned. However, for the planar fit coordinate rotation, one would want to know whether a sectorial fit is being employed (in complex terrain, most generally the 'plane' is not the same for different wind directions…, see, e.g., Stiperski and Rotach 2016), and over which period the 'planes' were fitted.

2) Homogeneous snow cover: is indeed crucial. Specially mentioned are (l. 156ff) periods with $T\_s > 1.5\ °C$ and periods with albedo <0.4. But surely, this is mentioned only to detail some special cases. What are the criteria for homogeneity in the first place?

3) Energy available for melt (eq. 1): First of all, this assessment neglects storage in the snowpack (if we have the sum of all the mentioned energy fluxes being non-zero, there is excess energy available (positive or negative) to heat/cool the snowpack, and if zero degrees should be reached (at the surface), this will result in snow melt. Second, eq. (1) assumes that the energy balance is closed. Of course, the EB should be closed at the surface, but it rarely is (and the authors show themselves – even If I do not quite understand what they do in Section 3.2.5 (see there) – that the EB is not closed (not at all). Even over benign surfaces, differences (to closure) are typically several tens of percent (60-90% or so). In complex terrain (as in the present case) the issue is more pronounced (Rotach et al. 2008) - because of the (local) inhomogeneity

(not only of the snow cover – also the terrain itself and hence turbulence) and the advection (also vertical – hence the importance of the coordinate rotation!). Typically, in complex terrain, we do have flux divergence (when we measure the turbulent fluxes at 3 m height, they do not correspond to the surface fluxes [which are those relevant to the EB]. See for example Nadeau et al. (2013), e.g. their Fig. 4, or Sfyri et al. 2018). Usually, it is thought that EB under-closure is due to either instrument uncertainty (must be under-capture, of course), missing processes (e.g., meso-scale quasi-steady circulations) or incomplete corrections /post-processing. Note that in complex terrain, we have, by definition, meso-scale circulations such as thermally driven slope flows (also katabatic winds are in their nature thermally driven flows) or dynamically driven flow modification. And these are associated with non-zero vertical wind (and hence vertical advection). All this leads to an often quite pronounced under-closure of the EB. Basically, then, when assessing the 'melt energy' in the way the authors do, it will be 'too large' (or at least an 'upper limit estimate'. If the EB would indeed be closed at this site (and 3m measurement height) – which you show is not the case – $Q_m$ would be the storage/melt energy. In any other case $Q_m$ would actually be smaller. Unfortunately, all the procedures to minimize the under closure are flow dependent – so one cannot simply ignore the 'corrections' (i.e. additional terms like advection and flux divergence terms).

4) L. 184: clustering. This clustering approach sounds interesting – but I think I am not the only one who first hears about it. As it is described, it is purely statistical (which is fair enough), but somehow one would want to know whether or not the different clusters produce different synoptic situations. Only when I checked the given reference (Theobald et al 2015) I saw to what degree different clusters correspond to different synoptic conditions. I suggest to make this very clear (not only 'was verified through manual analysis'... - whatever this means), but by explicitly referring to the 'figures below' (4, 5, 6) where these synoptic patterns can be discerned.

5) L. 256. Look-up tables. Apparently, more than 50% of the data is not measured but

statistically 'created'. I assume (but the authors will have to detail this) that the look-up tables correspond to some sort of multi-linear regression model. I do not understand what the '100 iterations' actually mean and how the authors used the available valuable measurements to create (train) the 'look-up tables' and an independent part of the data to verify (and calculate the RMSE's).

6) L. 278: ...mean daily net energy flux...: does this make any sense? Are the individual daily cycles of the available energy (or the averages, if you will) different for the different clusters (synoptic conditions that is)? Do the distributions of net daily energy overlap? And if so (what I assume), how strongly? To what degree are those means dependent on the abundance of each cluster? Before the 'climatological trends etc. are assessed, this should be analysed I detail for 2016/17.

7) L.364 (Tab 5) I note that for T2, T4 and T6 the rows sum up to 1.01 (which is probably a rounding error). But for T7 to 1.23. Something must be wrong here. This will also (potentially) change Fig 9.

8) L. 453. Energy balance closure: I don't understand what is being done here. In fact, eq. (1) is a manifestation of energy balance closure (all what doesn't balance goes into $Q_m$, i.e. 'melt' (and storage) - but see comment 3). Does this mean that the authors assessed the closure without $Q_m$ (i.e., by choosing the corresponding conditions, it is assumed that $Q_m$ is zero)? First, it should be noted that still storage can be an issue (e.g, if T<0°C the snow pack can cool, negative storage). Anyway, if so, the authors are basically saying that even under conditions of non-likely melt+storage, there are huge additional processes challenging the energy balance closure. Some were mentioned in comment 3) above. Advection, in particular (horizontal and vertical!).

Minor comments

l. 158 ...were considered not (?) to have heterogeneous.... Either 'not to have homogeneous' or 'considered to have heterogeneous..'

l. 220 to remove. . .. This is, in complex terrain, not to remove any errors (that is what Wilczac has originally derived it for); rather, it is to align the coordinate system with the surface.

l. 241 surely the signal has units..: <0.7mV?

l. 314 are considered to be transition types: based on what? These are static (statistical) fields, so how can the authors claim this?

l. 350 recent cold frontal passage: where do we know from? Above (l. 314), T3 was not among the 'transitional regimes'. . .).

l. 476 linear trends: I basically see 'no trend' for T1, T3, T4, T5. . ... Has any significance analysis (in a statistical sense) been performed?

l. 487 is shown to decrease: at least, I think the decrease should be put in context to the year-to-year variability.

l. 493 are these indices determined over the same time period (I hope so)? More importantly, how do they vary during this time period? The figure would gain a lot, if each dot would be complemented with an 'error bar' showing the variability of the respective index in the respective year.

l. 504 in the nineties: at least two also in the eighties.

l. 608 can indicate significant increases...: in fact, the authors seem to demonstrate a negative trend (in total energy provided to the snowpack). So, the indication might rather be a significant reduction in ablation (or a change, to be more cautious).

References

Bellaire et al: 2017, http://dx.doi.org/10.1016/j.coldregions.2017.09.013

Nadeau DF, Pardyjak ER, Higgins CW, Parlange MB: 2013, Similarity scaling over a steep alpine slope. Boundary-Layer Meteorol 147:401–419

Rotach MW, Stiperski I, Fuhrer O, Goger B., Gohm A, Obleitner F, Rau G, Sfyri E, Vergeiner J: 2017, Investigating Exchange Processes over Complex Topography: the Innsbruck-Box (i-Box), Bull Amer Meteorol Soc, 98, No 4, 787-805, doi: 10.1175/BAMS-D-15-00246.1

Rotach MW, Andretta M, Calanca P, Weigel AP, Weiss A: 2008, Turbulence characteristics and exchange mechanisms in highly complex terrain, Acta Geophysicae, 56 (1), 194-219. https://doi.org/10.2478/s11600-007-0043-1

Sfyri E, Rotach MW, Stiperski I, Bosveld FC, Obleitner F, Lehner M: 2018, Scalar flux similarity in the layer near the surface over mountainous terrain, Boundary-Layer Meteorol, 169 (1), 11–46, doi: 10.1007/s10546-018-0341-y.

Stiperski I, Rotach MW: 2016, On the measurement of turbulent fluxes over complex mountainous topography, Boundary-Layer Meteorol, 159, 97–121, DOI 10.1007/s10546-015-0103-z

---

## Referee Comment (RC2) · Anonymous Referee #2 · 13 May 2019

General Comment:

This manuscript tackles the interesting problem of linking the energetics of a marginal snowpack to large-scale atmospheric circulation using a synoptic weather type approach. Meteorological observations obtained from an automatic weather station over two winters (2016 and 2017), including direct measurements of the turbulent heat fluxes using an eddy covariance approach, form the basis to establish the long term synoptic controls on snowpack variability determined using ECMWF ERA-Interim data over a 39 year period. The key conclusion is that energy available for melt has decreased over the long term due to a reduction in the number of precipitation generating

cold fronts and associated preceding warm air advection.

The key conclusion is significant so two major comments are provided that reflect on the evidence provided to demonstrate this. In doing so, it needs to be acknowledged that the comments already provided by RC1 (Mathias Rotach) have been reflected on. Some of the key points are raised again, while others are not as they have already been carefully discussed.

Major comments:

1. Bormann et al. (2012) demonstrate that snow cover is decreasing in Australian alpine regions using MODIS data, which is consistent to some of the work cited in L84-L96, but the key finding in this manuscript is that energy available for melt has decreased. This finding is contrary to what one might expect in a warming world, where higher air temperatures typically lead to more energy being available for melt. However, it is argued in L582-L588 that reductions in precipitation (snowfall) might be causing the decrease in snow rather than changes in energy available for melt. If it is the latter and the authors wish to maintain that energy available for melt has decreased, it is the view of this reviewer that more evidence is still required. There is no evidence in the present manuscript to demonstrate that precipitation has actually decreased. One of the key issues is whether the proportion of precipitation that falls as rain rather than snow has changed over time. It could be argued that the modelling approach in this study does not adequately account for the feedback associated with warming and the associated changes in phase of precipitation. But perhaps more importantly, it is the method used to link the optimum number of synoptic weather types (L273-L282) to the mean daily net snowpack energy flux representative of each of the types that is the most tenuous. Looking at changes in frequency of the synoptic events over time seems like an interesting pursuit but I am not sure whether the energy balance values presented in this manuscript over two winters are adequate to build a representative snowpack energy flux for each weather type over 39 seasons. Reasons for this are given in the following three points.

[Figure]

2. RC1 raised the issue about how representative one site is for determining the energy available for melt for the entire alpine region. The answer to this question has actually been answered by some of the coauthors of the present manuscript in Bilash et al. (2019). It is clearly stated that "individual point measurements are unable to fully represent the variability in the snowpack across a catchment" and that "we show that recognizing and addressing this variability are particularly important for studies in marginal snow environments". This would seem to suggest that using a point based measurement to represent the energy status of the snowpack for the Australian Alps for the purpose of linking it to large-scale atmospheric circulation is unsatisfactory.

3. It could still be argued that a point based approach is a valuable first cut at the problem, but for this to be the case it is essential that the observational data and model are of the highest quality. The marginal snowpack being described appears to be particularly vulnerable to rain on snow events, and periods where snow disappears completely from the ground at some locations in the alpine catchment. It might be advisable to use a snow model that can account explicitly for the changes in temperature structure of the underlying soil and snowpack to enable a much more careful time series of snow depth and structure to be described, as well as to identify unequivocally when melt is occurring. This would allow the energy for melt to be explicitly described, as opposed to the energy balance of a cold winter snowpack, which is likely to have quite a different energy balance compared to conditions defined by melt (e.g. Cullen and Conway, 2015).

4. The post-processing and quality of the eddy correlation data has been raised by RC1, in particular the limitation of using look up tables to fill a large proportion of missing data. Importantly, it is unknown how much data during the prefrontal and frontal (periods of precipitation) are actually real, which is somewhat crucial given the importance placed on these weather types on both the energy status and precipitation. The question that begs to be asked in relation to this methodological issue is why depend on eddy correlation data to build a time series of energy exchanges over the

two winters? Would an approach used by Bilash et al. (2018) not be more suitable, which would be to use high quality eddy correlation data to support a bulk aerodynamic method? Even the use of a degree-day approach might allow the key findings of this manuscript to be more robustly tackled, which is whether the energy status (arguably governed to some extent by warming) or changes in precipitation are controlling the change in snow cover through variations in weather types over a long period (39 years). The suggestions by RC1 about the post-processing of the eddy covariance data are very important, and once these have been done it is recommended that the authors carefully consider whether another approach should be adopted to construct an energy balance over multiple years (e.g. to use a more sophisticated snow model, or apply a bulk aerodynamic and/or a degree-day modelling approach).

Other comments:

1. L43-44: It is stated that research related to synoptic influences on energy balance over marginal snowpacks are rare. To provide greater context for this, it might be useful to more carefully define marginal as there are a number of research articles linking energy exchanges over snow and synoptic conditions that the authors have not considered (e.g. Yarnal, 1984; Romolo et al., 2006; Käsmacher and Schneider, 2001; Matthews et al., 2015; Isaksen et al., 2016, Cullen et al., 2019).

2. L198-L288: No information about the calculation of QG is provided in the model description, despite being explicitly referred to in Section 3.2.3 and again in Section 4.1. As noted above, it might be worthwhile considering using a more comprehensive snow model that allows the evolution of the snowpack to be reconstructed and compared to observations. The approach to define whether a period is snow covered or not is described in Section 2.3 but from this it is hard to know just how much snow there is on the ground at any one time. Modelling the evolution of the snowpack and comparing that to observations would seem crucial given the emphasis on establishing the changes in the energetics of the snowpack over a longer temporal period.

3. Section 3.2; It could be useful to more carefully compare the energy balance results derived using the eddy covariance data in this study to those generated by Bilash et al. (2018) using a validated bulk aerodynamic approach. Monthly and entire cold season (winter) data could be compared prior to presenting it within a daily synoptic type framework. This suggestion stems from the fact that the presentation of the energy balance results in the current manuscript are framed differently to those described by Bilash et al. (2018), where it is stated that 80% of the energy is sourced from incoming longwave energy. It may be that the data are essentially the same but the way they are presented places emphasis on different components of the energy balance. For example, on L519 it is stated that net shortwave radiation contributes the largest amount of energy to the snowpack, which is a different message to Bilash et al. (2018).

4. L534-L537: It is suggested that "many prior works have not made the distinction" between shortwave and longwave radiation. Historically, this may have been the case but there are numerous examples of more recent research over snow and ice that have used similar radiometers to those used by the authors to carefully identify the importance of individual radiation terms. The importance of moisture and clouds has been focused on by a number of researchers working over mid-latitude glaciers and ice sheets.

Technical and specific corrections

L132: There is a need to be more explicit that two winters are used as the observational dataset, not to just state when the observations began. L324: K and L are not explicitly defined in the manuscript in the form presented. L334: Manual classification of cloud cover requires clarification.

References

Bilish, S. P., McGowan, H. A., and Callow, J. N.: Energy balance and snowmelt drivers of a marginal subalpine snowpack, Hydrological Processes, 32, 3837-3851, https://doi.org/10.1002/hyp.13293, 2018.

Bilish, S. P., Callow, J. N., McGrath, G. S., and McGowan, H. A.: Spatial controls on the distribution and dynamics of a marginal snowpack in the Australian Alps, Hydrological Processes, https://doi.org/10.1002/hyp.13435, 2019.

Bormann, K. J., McCabe, M. F., and Evans, J. P.: Satellite based observations for seasonal snow cover detection and characterisation in Australia, Remote Sensing of Environment, 123, 57–71, https://doi.org/10.1016/j.rse.2012.03.003, 2012.

Cullen, N. J. and Conway, J. P.: A 22-month record of surface meteorology and energy balance from the ablation zone of Brewster Glacier, New Zealand, Journal of Glaciology, 61, 931–946, https://doi:10.3189/2015JoG15J004, 2015.

Cullen, N.J., Gibson, P.B., Mölg, T., Conway, J., Sirguey, P., and Kingston, D. G.: The influence of weather systems in controlling mass balance in the Southern Alps of New Zealand. Journal of Geophysical Research: Atmospheres, 124, https://doi:10.1029/2018JD030052, 2019.

Isaksen, K., Nordli, Ø., Førland, E. J., Łupikasza, E., Eastwood, S. and Niedźwiedź, T.: Recent warming on Spitsbergen - Influence of atmospheric circulation and sea ice cover. Journal of Geophysical Research: Atmospheres, 121, 11,913–11,931. https://doi:10.1002/2016JD025606, 2016

Käsmacher, O. and Schneider, C.: An objective circulation pattern classification for the region of Svalbard. Geografiska Annaler: Series A, Physical Geography, 93, 259-271. https://doi:10.1111/j.1468-0459.2011.00431.x, 2011.

Matthews, T., Hodgkins R., Wilby R. L., Guðmundsson S., Pálsson F., Björnsson H., & Carr S.: Conditioning temperature‐index model parameters on synoptic weather types for glacier melt simulations. Hydrological Processes, 29, 1027-1045. https://doi.org/10.1002/hyp.10217, 2015.

Romolo, L., Prowse, T.D., Blair, D., Bonsal, B.R., & Martz, L.W.: The synoptic controls on hydrology in the upper reaches of the Peace River Basin. Part I: Snow Accumulation. Hydrological Processes, 20(19), 4097–4111. https://doi.org/10.1002/hyp.6421, 2006.

Yarnal, B.: Relationships between synoptic-scale atmospheric circulation and glacier mass balance in south-western Canada during the international hydrological decade, 1965–74. Journal of Glaciology, 30, 188–198, 1984.

---

## Author Comment (AC1) · 10 Jun 2019

The comment was uploaded in the form of a supplement:
https://www.the-cryosphere-discuss.net/tc-2019-48/tc-2019-48-AC1-supplement.pdf

---

## Author Comment (AC2) · 10 Jun 2019

This comment is a response to the review comments provided by the two reviewers of our article "Quantifying the impact of synoptic weather types, patterns, and trends on energy fluxes of a marginal snowpack".

We would like to thank the reviewers, Prof Mathias Rotach and another anonymous reviewer, for their time to thoroughly review our paper, give positive comments, and offer helpful suggestions on how to improve the paper.

The purpose of this comment is to provide an open response to the comments and suggestions of the reviewers, to outline how we propose to address these suggestions within a revised manuscript, or to justify our current approach in relation to specific suggestions or comments.

The overarching critical comments of the two reviewers and our proposed response is summarized below:

1.  Data processing and analysis methods for the energy balance data (R1CI1, R1MC1, R1MC3, R1MC5, R1MC6, R1MC8, R1C8, R2MC4, R2C2)
    - We tested the approach suggested by R1, Prof Mathias Rotach, and reprocessing Qh and Qe data using the double coordinate rotation method to better address the complex terrain of the site and associated potential impacts from mesoscale and thermodynamic processes, which improved the accuracy and will implement this change.
    - The above mentioned reprocessing reduced the amount of overall flux data needing to be gap-filled, however, we have updated to use a multi-variate regression approach supported by the random forest algorithm in order to reduce uncertainty in gap-filled data further.
    - Add a snowpack storage term to eqn.1 to address the non-closure of the energy balance system and discuss why a lack of system closure is likely in environments such as the Snowy Mountains.
2.  The representativeness of the Eddy Covariance site (R1CI1, R2MC2)
    - We add detail around why this single-station study is actually quite representative of this study area. We provide more detail that the single EC site is representative of a significant portion of the Australian snow-covered area, which exists in a relatively narrow elevation band between 1400-2200m (site at ~1830m), over a limited number vegetation types of which the site footprint included the majority of these and in representative proportions.

Below we respond in detail. The reviewers' comments are in black text, and we identify them with the prefix R1 for Prof Mathias Rotach and R2 for the other anonymous reviewer, followed by a critical, major, general or technical comment and number suffix ("CIx", "MCx", "Cx", where x is the comment number). Our response of either our proposed course of action to address the point or justification of the current approach is in blue text. This includes how we have proposed to response in the manuscript.

Regards,

Andrew Schwarz and on behalf of co-authors

**Mathias Rotach (Referee 1)**

mathias.rotach@uibk.ac.at

General considerations

In this paper, the authors set out to investigate the impact of the synoptic flow conditions (and their changes over the years) to available energy for a snow pack in South East Australia. This is, first of all, a very valuable undertaking and adds to providing additional scientific understanding on potential causes and mechanisms of impact (going beyond the simplistic global change → warmer → more snow melt). For this purpose, the authors use surface data from one energy balance station (situated on the snow pack) in connection with ERA-Interim reanalysis data for the synoptic situation. It seems to me that in both these data sources, there are conceptual problems that need to be addressed before the paper can be recommended for publication. I usually, when preparing my review (i.e., reading the paper) list 'major' and 'minor' issues separately (this is what can be found below). Still, I add those two critical issues separately – even if some of the 'major' (and even some of the minor) comments address the same topics.

I have tried to give some references for specific points raised – and they often happen to be from my own work. This is not 'to make the authors cite my papers' – it is just that it is the quickest way to get this put together. Often, there are other, equally suited papers around for the same issue.

Response: We thank Reviewer 1, Prof Mathias Rotach, for his time reviewing this paper and for his positive comment describing the paper as *"a very valuable undertaking and adds to providing additional scientific understanding on potential causes and mechanisms of impact"*. We address the concerns of the reviewer where highlighted as critical, major, and minor comments/suggestions.

Critical issues

**R1CI1:** The authors use one surface EB station in an area of large spatial inhomogeneity, and derive all their results from this one site (2 years ['seasons'] of measurements). The individual contributions to the EB are then attributed to each synoptic type and further used in a climatological study. All this, basically prompts the questions: i) how representative is this one site for any larger area? Is the very local measurement representing the [energy distribution] in the 'Snowy Mountains' to any degree? ii) Are the energy estimates accurate enough to draw any firm conclusions? As for ii) the authors take most of the necessary corrections etc. to measurements appropriately into account and also apply a quality assurance procedure. However, this leads to the necessity to basically 'produce' 50% of the data (and I am not convinced that a lookup table is the best solution to do this, if necessary, see below) with a claimed uncertainty of some 21 W/m2 (latent heat flux) and 34 W/m2 (sensible heat flux). Summed over the day this amounts to some 1.8 (2.9) MJ/m2 (and day). Looking at Fig. 10 reveals that this is more for both Q_e and Q_H than the difference between the synoptic pattern with the largest and the smallest contribution to the EB. So, this triggers two consequences: First, the accuracy of the measurements has to be increased (which means not to use the Campbell standard post processing, taking into account (estimating) the uncertainty due to assumptions which are not particularly good (such as 'constant fluxes', such as planar fit coordinate rotation with only one plane', etc. – see some of the 'major comments'). Second, the estimation of missing data has to become better (I am pretty sure that much better accuracy can be obtained by using a sophisticated statistical model), and finally, the uncertainty of the estimates has to be taken into account when discussing the results (e.g. Fig. 10). For example, this 'bar plot' could include error bars.

Concerning the representativeness question (i), I can only say, that our own measurements of contributions to the EB in complex terrain (e.g. Rotach et al. 2017), yields huge differences in all the contributions (with the exception of the longwave balance) from site to site, at kilometre scale. We haven't investigated the impact on daily means (or sums), but looking at daily cycles shows that they can be substantial (several 100 W/m2 in the maximum and several hours of difference when the maximum occurs, i.e. when the fluxes start to 'go back'). I have no useful suggestion, how to actually estimate the potential attribution error that might be introduced when using only one site (based on published results, that is) – but at least, I think it must be discussed (maybe Bellaire et al. 2017 provide some hints on orders of magnitude).

We agree that single-site studies will always face this challenge, but can still make a very valuable contribution to advancing knowledge. While we accept that ideally there are multiple sites to address these issues, this specific site is actually highly representative of the Australian Alps region as a whole and was situated at its location for this reason. As Bilish et al. (2018, p. 3839) stated (on a collocated site), this site "*has features representative of a key part of the Australian Alps region. The two predominant vegetation types in the catchment, grassland/alpine bog and eucalypt woodland, are also the two most common types in the broader region, together representing 47% of the total area above 1400-m elevation*". Our site is also positioned at ~1830 m, which falls towards the middle of the 1400-2228 m snow zone in the Australian Alps allowing for measurements that apply for a broad range of elevations. As such, we strongly argue that this single-site study can actually be considered to be representative of this study region, particularly when considering that multi-site studies in many alpine areas are attempting to represent area that are orders of magnitude larger and cover more significant elevation-range and biomes.

Overall, the average size of the 90% flux footprint for the two (snow) seasons of eddy covariance measurements was 108 m, which equates to a measured area of 36,856 $m^2$ that includes many characteristics of the broader region. Alpine grassland/bog accounts for ~70% of the measured area within the 90% footprint with *Eucalyptus pauciflora* woodland comprising the other 30%. Gellie (2005) showed that the *E. pauciflora* woodland was present in five of the fifteen dominant vegetation formations that covers 57% of area within the broader region while Alpine grassland/bog (including herbfields) accounts for another 8%. We acknowledge that there will be some uncertainty when applying the energy balance to the wider area, however, this should be reduced as the measurements made include surface types common in the wider region.

Thank you for the helpful suggestions on how to improve our data quality and better communication of data uncertainty. We agree that the gap-filling method originally used on the data can be improved, please see our response to R1MC5 for our proposed solution to this problem.

**Manuscript change**: Clarify the specific details on how representative the site is of the broader region within the study setting, adding some of the above information to justify this single-station study.

In order to increase the accuracy of the measurements, we suggest the use of a double coordinate rotation system instead of the original planar fit due to the surrounding topography of the site and associated flow. Please see comment response to R1MC3, R1MC5 for more detail.

**R1CI2:** My second concern is the treatment of 'days' as an entity. Each day is identified with a 'synoptic pattern' (which is determined based on mean daily values of SLP, temperature, humidity, wind, etc.). While I can see how statistically this procedure yields a synoptic pattern for each day, I do not understand, why those patterns should always start at midnight and last for a day. Looking at the patterns shows that they constitute 'snapshots' of a dynamic development (e.g. the T7-T5-T3 sequence (Fig. 9) actually corresponding to an eastward moving trough to the south of the area of interest). This means that those patterns are only 'statistical entities' with the actual position of the trough on a given day (00.00 to 24.00 o'clock) being 'closest' to one of the patterns. If the trough had built (or moved in) at 12.00 o'clock (rather than at midnight and 'stayed there' for 24 hrs) as manifested in T7, the actual pressure distribution (on day two, say) would likely be somewhere between T7 and T5 – with possibly quite drastic consequences for the local pressure (geopotential) distribution (e.g. Fig. 4) and hence advection pattern. This, in turn, will (potentially) affect large parts of the explanations (e.g., the WAA and CAA as discussed in Section 3.1.3). Can the authors 'defend' their choice of 24 hr time segments in some more detail and comment on the potential consequences (day-to-variability within a given pattern)?

Thank you for the suggestion to improve the justification of "days" as the analysis period. Any analysis of this nature will require a time period to be defined, and we select days as it was a sensible mid-point that allowed for multi-decadal synoptic type differentiation that could be related to shorter-term surface energy balance characteristics and long-term climatological trends. The time-step chosen allows for the impact of synoptic systems on seasonal or decadal time-scales to be studied while still accurately describing the energy being delivered to the landscape. One of the primary issues with using a smaller time step would be the development and comparison of synoptic types with fundamentally different radiation flux characteristics due to the diurnal radiation cycle. For those reasons, we adopted the approach of other literature in similar types of studies (Chubb

et al., 2011; Neale and Fitzharris, 1997; Theobald et al., 2016; Theobald et al., 2015) and used days for the synoptic analysis.

**Manuscript change**: add section to methods to clarify that we use days, similar to Chubb et al., 2011; Neale and Fitzharris, 1997; Theobald et al., 2016; Theobald et al., 2015.

Major comments

**R1MC1:** L. 131: It is good to provide the information on the instrumentation. It is also necessary, however, to provide information on data post-processing. Later (l. 217 ff), I see most of it is mentioned. However, for the planar fit coordinate rotation, one would want to know whether a sectorial fit is being employed (in complex terrain, most generally the 'plane' is not the same for different wind directions..., see, e.g., Stiperski and Rotach 2016), and over which period the 'planes' were fitted.

**Manuscript change**: we have taken on-board this suggestion, and will implement this with the new processing procedures used  See R1MC3 and R1MC5).

For reference/record we had originally used Planar fit coordinate rotation with a sectorial planar fit over four sectors calculated from 30 minute block averages from 211 days over the two winters for a total of 10,128 half-hour periods.

**R1MC2:** Homogeneous snow cover: is indeed crucial. Specially mentioned are (l. 156ff) periods with $T_s > 1.5$ °C and periods with albedo <0.4. But surely, this is mentioned only to detail some special cases. What are the criteria for homogeneity in the first place?

We agree with this comment. By using our snow depth, surface temperature, and albedo data, we included only data considered to be a homogeneous snowpack, with no grass or bush protruding from the snow surface with the exception of the more distant surrounding Eucalyptus trees.

**Manuscript change**: Add more detail on what was considered to be homogenous snow cover during the measurement period.

**R1MC3:** Energy available for melt (eq. 1): First of all, this assessment neglects storage in the snowpack (if we have the sum of all the mentioned energy fluxes being non-zero, there is excess energy available (positive or negative) to heat/cool the snowpack, and if zero degrees should be reached (at the surface), this will result in snow melt. Second, eq. (1) assumes that the energy balance is closed. Of course, the EB should be closed at the surface, but it rarely is (and the authors show themselves – even If I do not quite understand what they do in Section 3.2.5 (see there) – that the EB is not closed (not at all). Even over benign surfaces, differences (to closure) are typically several tens of percent (60-90% or so). In complex terrain (as in the present case) the issue is more pronounced (Rotach et al. 2008) - because of the (local) inhomogeneity (not only of the snow cover – also the terrain itself and hence turbulence) and the advection (also vertical – hence the importance of the coordinate rotation!). Typically, in complex terrain, we do have flux divergence (when we measure the turbulent fluxes at 3 m height, they do not correspond to the surface fluxes [which are those relevant to the EB]. See for example Nadeau et al. (2013), e.g. their Fig. 4, or Sfyri et al. 2018). Usually, it is thought that EB under-closure is due to either instrument uncertainty (must be under-capture, of course), missing processes (e.g., meso-scale quasi-steady circulations) or incomplete corrections /post-processing. Note that in complex terrain, we have, by definition, meso-scale circulations such as thermally driven slope flows (also katabatic winds are in their nature thermally driven flows) or dynamically driven flow modification. And these are associated with non-zero vertical wind (and hence vertical advection). All this leads to an often quite pronounced under-closure of the EB. Basically, then, when assessing the 'melt energy' in the way the authors do, it will be 'too large' (or at least an 'upper limit estimate'. If the EB would indeed be closed at this site (and 3m measurement height) – which you show is not the case – $Q_m$ would be the storage/melt energy. In any other case $Q_m$ would actually be smaller. Unfortunately, all the procedures to minimize the under closure are flow dependent – so one cannot simply ignore the 'corrections' (i.e. additional terms like advection and flux divergence terms).

We agree that the eq.1 needs to be updated to include a snowpack storage term as the other terms of the equation will not equal zero and complete energy balance closure at energy balance sites is exceedingly rare, if not impossible.

After considering the reference provided (Stiperski and Rotach, 2016) and re-processing our data, we believe that double rotation is a more accurate approach for our energy balance budget as our site, surrounded by undulating topography without a uniform slope and is subject to anabatic and katabatic flow. The correction in coordinate rotation will allow for better estimation of the sensible and latent heat fluxes at the site and will correct for the effects of local advection, which may be responsible for producing an overestimate of $Q_m$. In addition, the interpretation of synoptic-scale influences requires that each synoptic type be treated as unique and using a planar fit rotation system assumes a mean climatological state that is present over seasons and derived during a variety of synoptic conditions. Therefore, it seems that planar fit is optimal for the determination and analysis of longer period energy budgets while double rotation would be a better option to allow for the accounting of the effects of synoptic systems and their potential mesoscale responses.

When testing with the double rotation coordinate rotation method, we have determined that the total $Q_h$ value is 33% higher and $Q_e$ value is 13% lower during the two winter periods. The double rotation method also reduced the amount of $Q_h$ data that needed to be gap-filled from 49% to 30% while the amount of gap-filled $Q_e$ data had a minor increase by 2% (53-55%), which will result in less overall measurement uncertainty from gap-filling procedures (see response to that suggestion below in R1MC5).

**Manuscript change**: Update data using double coordinate rotation method and recalculate synoptic type fluxes and climatology based on new values.

**R1MC4:** L. 184: clustering. This clustering approach sounds interesting – but I think I am not the only one who first hears about it. As it is described, it is purely statistical (which is fair enough), but somehow one would want to know whether or not the different clusters produce different synoptic situations. Only when I checked the given reference (Theobald et al 2015) I saw to what degree different clusters correspond to different synoptic conditions. I suggest to make this very clear (not only 'was verified through manual analysis'... - whatever this means), but by explicitly referring to the 'figures below' (4, 5, 6) where these synoptic patterns can be discerned.

Thank you for this suggestion, you are correct. We use the k-means clustering and cut-off the number of clusters needed to accurately describe the data using the inflection method, as is the common approach. The process is purely statistical and we use a rigorous approach to the number of input variables and capturing data through the atmosphere identical to Theobald et al. (2015). We then classified each day using one of the "n" clusters. To make this a bit less complicated to the reader, we use a manual approach to 'name' the synoptic types and link them to atmospheric processes that make sense. While different numbers of clusters will provide slightly different synoptic characteristics, there is a point when adding more clusters doesn't affect the ability to accurately classify a synoptic day, which is where the inflection point in Figure 3 lies (7 clusters/synoptic types).

**Manuscript change**: Add further description the synoptic typing/clustering methods.

**R1MC5:** L. 256. Look-up tables. Apparently, more than 50% of the data is not measured but statistically 'created'. I assume (but the authors will have to detail this) that the look-up tables correspond to some sort of multi-linear regression model. I do not understand what the '100 iterations' actually mean and how the authors used the available valuable measurements to create (train) the 'look-up tables' and an independent part of the data to verify (and calculate the RMSE's).

You are correct that the look-up tables used were created using a multi-variable linear regression model that took all known $Q_h$ and $Q_e$ fluxes and stored the associated stability, wind speed, relative humidity, and surface-atmospheric temperature difference in a multi-dimensional matrix. When gaps were encountered in $Q_h$ or $Q_e$, the associated variables were used to 'look up' the nearest value in the matrix. An algorithm was created that would remove a random section of $Q_h$ and $Q_e$ data from the existing values, construct a look-up table using the remaining values, gap-fill the missing data, and then compare to the original values to obtain RMSE. As the removal of the data was randomized, it was considered necessary to run the test multiple times ('100 iterations') to ensure that the look-up tables performed similarly in all situations when different subsets of the data were

removed. A mean RMSE was calculated using the RMSE from each of the 100 iterations to describe the gap-filling method's overall performance.

Although the initial approach to gap-filling the data was to use methods well documented by other papers, we have performed analysis on additional gap-filling methods since receiving the reviewers' comments and have decided that a more sophisticated statistical model, as suggested in R1CI1, is indeed needed to limit the uncertainty in the gap-filled measurements (reducing this from around 50% to 30%). The new method (a multi-variate regression model) will be used to reduce uncertainties in the 30-minute block averages and reduce any compound uncertainties when deriving the estimated energy flux climatology. Preliminary testing using the original data and new gap-filling method indicates reductions in RMSE from 33.9 W/m2 to 15.69 W/m2 for Qh and from 20.9 W/m2 to 11.71 W/m2 for Qe accounting for a significant reduction in uncertainty.

**Manuscript change**: Implement a multi-variate regression model supported by the random forest algorithm (Breiman, 2001) to gap-fill the Qh and Qe data with less uncertainty.

**R1MC6:** L. 278: *...mean daily net energy flux...*: does this make any sense? Are the individual daily cycles of the available energy (or the averages, if you will) different for the different clusters (synoptic conditions that is)? Do the distributions of net daily energy overlap? And if so (what I assume), how strongly? To what degree are those means dependent on the abundance of each cluster? Before the 'climatological trends etc. are assessed, this should be analysed I detail for 2016/17.

We agree with you that this topic needs to be revisited as using means to compare energy flux characteristics of each synoptic type does not give information regarding the variation of the energy flux associated with each type and that the mean may be sensitive to outliers. As such, we suggest recalculation of the daily net energy fluxes for each synoptic type using the five number summary (sample minimum, first quartile, median, third quartile, and sample maximum) to better depict the characteristics of each synoptic day's energy flux.

**Manuscript change**: Replace analysis on mean daily values for each synoptic types with more informational statistics to better describe the characteristics (and variability) of each synoptic type. Change Tables 2 & 7 as well as Figures 10, 12, 14, 15, & 16 to reflect the new statistics used.

**R1MC7:** L.364 (Tab 5) I note that for T2, T4 and T6 the rows sum up to 1.01 (which is probably a rounding error). But for T7 to 1.23. Something must be wrong here. This will also (potentially) change Fig 9.

We thank you for this comment and have been able to track down the problems with the transition table values. T2, T4, and T6 are, indeed, rounding errors and T7 had an accidentally duplicated value that had been copied from the T7 -> T3 transition.

**Manuscript change**: Change T7->T2 value on Table 5 to 0.00 and fix rounding errors on T2, T4, and T6.

**R1MC8:** L. 453. Energy balance closure: I don't understand what is being done here. In fact, eq. (1) is a manifestation of energy balance closure (all what doesn't balance goes into Q_m, i.e. 'melt' (and storage) - but see comment 3). Does this mean that the authors assessed the closure without Q_m (i.e., by choosing the corresponding conditions, it is assumed that Q_m is zero)? First, it should be noted that still storage can be an issue (e.g, if T<0∘C the snow pack can cool, negative storage). Anyway, if so, the authors are basically saying that even under conditions of non-likely melt+storage, there are huge additional processes challenging the energy balance closure. Some were mentioned in comment 3) above. Advection, in particular (horizontal and vertical!).

We agree with you that Qm would be the residual energy term when discussing closure of the energy balance system. An estimate of system energy balance closure was attempted using periods that were deemed as being least likely to have significant melt occurring within the snowpack. By definition, however, the marginal snowpack of the Snowy Mountains is nearly isothermal for the entirety of the season and periods without melt are rare. These conditions make the calculation of energy balance closure with the goal of a complete closure at least difficult if not impossible as we suggested on L458, "*Inability to account for nearly constant internal*

*snowpack melt processes has likely resulted in the low closure ratio and high variability in calculated energy balance closure.*"

**Manuscript change**: Remove calculated energy balance closure values in section 3.2.5 and, instead, discuss why calculations of energy balance closure are impractical given the snowpack properties.

Minor comments

**R1C1:** l. 158 *...were considered not (?) to have heterogeneous...*. Either 'not to have homogeneous' or 'considered to have heterogeneous.'

**Manuscript change**: Change L158 to "… were considered to have heterogeneous snow cover…".

**R1C2:** l. 220 *to remove...*. This is, in complex terrain, not to remove any errors (that is what Wilczac has originally derived it for); rather, it is to align the coordinate system with the surface.

We agree with the point that you are making regarding the difference between coordinate rotation applications in complex terrain vs level terrain. While we acknowledge that a significant portion of energy balance studies that occur within complex terrain need to correct for a sloped ground surface beneath the instrumentation, the Pipers Creek site is relatively level with an estimated slope of approximately 2-3 degrees at the instrumentation site. As such, the Wilczac 2001 corrections were applied with the primary goal of reducing levelling errors at the site.

**Manuscript change**: Include additional details on site aspect and slope in section 2.1 and add 1-2 sentences clarifying our reasons to section 2.6 discussing the differences of complex terrain vs instrument levelling coordinate rotation and the applicability of both to our site.

**R1C3:** l. 241 surely the signal has units..: <0.7mV?

Gas analyser signal strength for the Campbell Scientific EC150 is calculated as the fraction of emitted vs received infrared beam strength, which results in a unitless number.

**R1C4:** l. 314 are considered to be transition types: based on what? These are static (statistical) fields, so how can the authors claim this?

**Manuscript change**: add clarification regarding the definition of 'transition types'.

**R1C5:** l. 350 recent cold frontal passage: where do we know from? Above (l. 314), T3 was not among the 'transitional regimes'...).

**Manuscript change**: Remove phrase 'from recent cold front passage' from sentence on L350.

**R1C6:** l. 476 linear trends: I basically see 'no trend' for T1, T3, T4, T5..... Has any significance analysis (in a statistical sense) been performed?

**Manuscript change**: Remove discussion of frequency changes for types T1, T3, T4, and T5 as there is no significant change in frequency for the types.

**R1C7:** l. 487 is shown to decrease: at least, I think the decrease should be put in context to the year-to-year variability.

**Manuscript change**: 1-2 sentences will be added discussing the year-to-year variability of the calculated energy flux.

**R1C8:** l. 493 are these indices determined over the same time period (I hope so)? More importantly, how do they vary during this time period? The figure would gain a lot, if each dot would be complemented with an 'error bar' showing the variability of the respective index in the respective year.

The correlations between snowpack season and teleconnection indices are derived from the same time period as the climatology (1979-2017).

**Manuscript change**: Error bars will be added to Figure 15 to illustrate seasonal variation in teleconnection phase and discussion of variation will be added in section 3.2.2.

**R1C9:** l. 504 in the nineties: at least two also in the eighties.

**Manuscript change**: Include reference to two years within the 1980s with reduced energy flux (though, this may change with the reprocessing of data).

**R1C10:** l. 608 can indicate significant increases...: in fact, the authors seem to demonstrate a negative trend (in total energy provided to the snowpack). So, the indication might rather be a significant reduction in ablation (or a change, to be more cautious).

**Manuscript change**: Change wording of L608 to "…should be closely monitored as changes can indicate significant changes in snowpack ablation."

**References Provided by Reviewer 1:**

Bellaire et al: 2017, http://dx.doi.org/10.1016/j.coldregions.2017.09.013

Nadeau DF, Pardyjak ER, Higgins CW, Parlange MB: 2013, Similarity scaling over a steep alpine slope. Boundary-Layer Meteorol 147:401–419

Rotach MW, Stiperski I, Fuhrer O, Goger B., Gohm A, Obleitner F, Rau G, Sfyri E, Vergeiner J: 2017, Investigating Exchange Processes over Complex Topography: the Innsbruck-Box (i-Box), Bull Amer Meteorol Soc, 98, No 4, 787-805, doi: 10.1175/BAMS-D-15-00246.1

Rotach MW, Andretta M, Calanca P, Weigel AP, Weiss A: 2008, Turbulence characteristics and exchange mechanisms in highly complex terrain, Acta Geophysicae, 56 (1), 194-219. https://doi.org/10.2478/s11600-007-0043-1

Sfyri E, Rotach MW, Stiperski I, Bosveld FC, Obleitner F, Lehner M: 2018, Scalar flux similarity in the layer near the surface over mountainous terrain, Boundary-Layer Meteorol, 169 (1), 11–46, doi: 10.1007/s10546-018-0341-y.

Stiperski I, Rotach MW: 2016, On the measurement of turbulent fluxes over complex mountainous topography, Boundary-Layer Meteorol, 159, 97–121, DOI
10.1007/s10546-015-0103-z

**Anonymous Referee #2**

General Comment:
This manuscript tackles the interesting problem of linking the energetics of a marginal snowpack to large-scale atmospheric circulation using a synoptic weather type approach. Meteorological observations obtained from an automatic weather station over two winters (2016 and 2017), including direct measurements of the turbulent heat fluxes using an eddy covariance approach, form the basis to establish the long term synoptic controls on snowpack variability determined using ECMWF ERA-Interim data over a 39 year period. The key conclusion is that energy available for melt has decreased over the long term due to a reduction in the number of precipitation generating cold fronts and associated preceding warm air advection.

The key conclusion is significant so two major comments are provided that reflect on the evidence provided to demonstrate this. In doing so, it needs to be acknowledged that the comments already provided by RC1 (Mathias Rotach) have been reflected on. Some of the key points are raised again, while others are not as they have already been carefully discussed.

Major comments:
**R2MC1:** Bormann et al. (2012) demonstrate that snow cover is decreasing in Australian alpine regions using MODIS data, which is consistent to some of the work cited in L84-L96, but the key finding in this manuscript is that energy available for melt has decreased. This finding is contrary to what one might expect in a warming world, where higher air temperatures typically lead to more energy being available for melt. However, it is argued in L582-L588 that reductions in precipitation (snowfall) might be causing the decrease in snow rather than changes in energy available for melt. If it is the latter and the authors wish to maintain that energy available for melt has decreased, it is the view of this reviewer that more evidence is still required. There is no evidence in the present manuscript to demonstrate that precipitation has actually decreased. One of the key issues is whether the proportion of precipitation that falls as rain rather than snow has changed over time. It could be argued that the modelling approach in this study does not adequately account for the feedback associated with warming and the associated changes in phase of precipitation. But perhaps more importantly, it is the method used to link the optimum number of synoptic weather types (L273-L282) to the mean daily net snowpack energy flux representative of each of the types that is the most tenuous. Looking at changes in frequency of the synoptic events over time seems like an interesting pursuit but I am not sure whether the energy balance values presented in this manuscript over two winters are adequate to build a representative snowpack energy flux for each weather type over 39 seasons. Reasons for this are given in the following three points.

You raise a good point as our work shows a decline in the energy available for snowmelt over the 39 year period, but also discusses reductions in precipitation that may be the cause of reduced snowcover. Previous work by some of the co-authors of this manuscript does show several trends in precipitation in the region including reductions during spring and autumn and "*an overall decline in cool-season precipitation amount and frequency of precipitation days ≥ 1 mm and ≥ 10mm*" (Theobald et al., 2016, p. 444). Additionally, the previous work discusses an expected decrease in precipitation associated with cold frontal systems, but also an increase in summer precipitation. So, while our work doesn't explicitly identify a reduction in precipitation, we were attempting to relate the research in this manuscript to the larger body of work in the region by discussing it in terms of reduced precipitation/snowpack.

We agree that using two years of snowpack energy flux data to calculate a 39 year climatology can be problematic if data is of low quality or there is concern about the representativeness of the data/site. This uncertainty in the past climatology is the reason that it was labelled as 'estimated mean seasonal' snowpack energy flux. Many of the suggestions of both reviewers have helped us to better clarify the representativeness of the site, and to use method to improve the accuracy of the data used to create the energy estimates for the 39 year climatology, though the approach to reduce uncertainty by applying different correction and gap-filling strategies (esp see response to R1MC3 and R1MC5). There is some expected uncertainty in the calculations of the estimated climatology as this was never actually measured, but the calculations give a good estimate of the flux of each season, based on fluxes derived from the synoptic type associated with the fluxes data during the period that this was recorded.

**Manuscript change**: Rewrite sections of the manuscript that directly discuss precipitation generating systems and synoptic types. Improve how we more clearly acknowledge measurement based uncertainty in estimated climatology and energy flux.

**R2MC2:** RC1 raised the issue about how representative one site is for determining the energy available for melt for the entire alpine region. The answer to this question has actually been answered by some of the coauthors of the present manuscript in Bilash et al. (2019). It is clearly stated that "individual point measurements are unable to fully represent the variability in the snowpack across a catchment" and that "we show that recognizing and addressing this variability are particularly important for studies in marginal snow environments". This would seem to suggest that using a point based measurement to represent the energy status of the snowpack for the Australian Alps for the purpose of linking it to large-scale atmospheric circulation is unsatisfactory.

See our response to R1C1, on site representation. It is worth clarifying that the Bilish et al. (2019) manuscript was discussing this in relation to a detailed study of snow depth and snow water equivalent dynamics within this same alpine catchment. As R1 identified net LW will be less subject to issues with specific site, and the work of Bilish (2018), shows that this is ~80% of positive energy input to the snowpack in this area measurements. The average size of the 90% flux footprint for the two seasons of eddy covariance measurements was 108 m, which equates to a measured area of 36,856 m$^2$ and includes many characteristics of the broader region biomes (please see response to R1CI1 for additional details). We accept that a multiple-site study will always be superior to a single-site study, but make strong argument that single-site studies are still valuable and better to be published to provide this data and insight to the research community, and that this specific site is actually quite representative.

**Manuscript change**: Add discussion of representativeness and variability as outlined in response to R1CI1.

**R2MC3:** It could still be argued that a point based approach is a valuable first cut at the problem, but for this to be the case it is essential that the observational data and model are of the highest quality. The marginal snowpack being described appears to be particularly vulnerable to rain on snow events, and periods where snow disappears completely from the ground at some locations in the alpine catchment. It might be advisable to use a snow model that can account explicitly for the changes in temperature structure of the underlying soil and snowpack to enable a much more careful time series of snow depth and structure to be described, as well as to identify unequivocally when melt is occurring. This would allow the energy for melt to be explicitly described, as opposed to the energy balance of a cold winter snowpack, which is likely to have quite a different energy balance compared to conditions defined by melt (e.g. Cullen and Conway, 2015).

We agree that this is a valuable first-pass at the problem and we currently have forthcoming manuscripts that will address much of the issue of snowpack energy balance, including the finer scale dynamics, in the Snowy Mountains in higher detail. In large part, we take these comments on board as being outside the scope of this paper, but critical issues to address in detail within forthcoming work as part of this research program.

While the snowpack can be susceptible to rain-on-snow events, analysis by Bilish et al. (2018) agrees with ours that little energy is contributed to the snowpack in this manner and the periods effected by significant rain-on-snow events that resulted in heterogeneous snow cover have been removed using our methods outlined on L156. However, rain-on-snow events are an area of ongoing research interest within our research group, and an area that warrants a more detailed and specific approach to this topic – detail that is beyond the scope of what is achievable within this paper.

While a useful pursuit and considered for possible future expansion on the existing body of knowledge on snowpacks in the Snowy Mountains region, we consider the finer scale dynamics of the snowpack and its modelling outside the scope of this work as we seek only to analyse the differences in snowpack energy balance under different synoptic regimes. At present, in our opinion, there is no suitable snowmelt model for our field site or the Australian Alps, however, we hope that our work will inform the development of such a model.

As mentioned in our response to R1MC3, we will be updating our snowpack energy balance model to include a snowpack storage term to better represent the energy balance and under-closure present when measuring these types of systems.

**R2MC4:** The post-processing and quality of the eddy correlation data has been raised byRC1, in particular the limitation of using look up tables to fill a large proportion of missing data. Importantly, it is unknown how much data during the prefrontal and frontal (periods of precipitation) are actually real, which is somewhat crucial given the importance placed on these weather types on both the energy status and precipitation. The question that begs to be asked in relation to this methodological issue is why depend on eddy correlation data to build a time series of energy exchanges over the two winters? Would an approach used by Bilash et al. (2018) not be more suitable, which would be to use high quality eddy correlation data to support a bulk aerodynamic method? Even the use of a degree-day approach might allow the key findings of this manuscript to be more robustly tackled, which is whether the energy status (arguably governed to some extent by warming) or changes in precipitation are controlling the change in snow cover through variations in weather types over a long period (39 years). The suggestions by RC1 about the post-processing of the eddy covariance data are very important, and once these have been done it is recommended that the authors carefully consider whether another approach should be adopted to construct an energy balance over multiple years (e.g. to use a more sophisticated snow model, or apply a bulk aerodynamic and/or a degree-day modelling approach).

Thank you for this suggestion, and see our detailed response to how we have reprocessed the data using the suggestions of R1 and improved data quality and reduced uncertainty. Despite the challenges in the eddy covariance method, the data obtained is still of high quality when the proper corrections are made (Stiperski and Rotach, 2016). Bilish et al. (2018, p. 3847) states that both methods can be problematic during stable atmospheric profiles as fluxes can decouple from the surface "*undermining the basis of the bulk aerodynamic method*" and "*EC flux measurements may become unrepresentative*" during periods of stability. However, the paper also mentions a couple other drawbacks of using the bulk aerodynamic approach: 1) that "*there is considerable uncertainty in the bulk aerodynamic method and its parameterizations*" (p. 3847), and 2) that there can be errors in energy balance calculation due to different footprint sizes of turbulent and non-turbulent fluxes. A degree-day modelling approach can be useful in a climatology, however, we believe that complete measurement of energy balance fluxes offers a more detailed analysis of the energy balance system and using the degree-day approach would act to obscure more precise results. For these reasons, we believe the continued use of the eddy covariance method is appropriate for our work – it is universally accepted as the gold standard for direct measurement of surface-atmosphere energy exchanges and is appropriate in mountainous terrain (Hammerle et al., 2007; Hiller et al., 2008).

As discussed in our responses to R2MC1 and R2C2, the aim of this paper is simply to identify the atmosphere-snowpack energy flux trends and not to examine snow cover change. This is the focus of forthcoming work devoted to this in the specific detail that work requires, and these comments are useful context for how we proceed with the analysis within that work.

Other comments:
**R2C1:** L43-44: It is stated that research related to synoptic influences on energy balance over marginal snowpacks are rare. To provide greater context for this, it might be useful to more carefully define marginal as there are a number of research articles linking energy exchanges over snow and synoptic conditions that the authors have not considered (e.g. Yarnal, 1984; Romolo et al., 2006; Käsmacher and Schneider, 2001; Matthews et al., 2015; Isaksen et al., 2016, Cullen et al., 2019).

**Manuscript change**: 1-2 sentences of additional detail defining 'marginal' will be added to section 1.1 to provide better context for our use of the term.

**R2C2:** L198-L288: No information about the calculation of QG is provided in the model description, despite being explicitly referred to in Section 3.2.3 and again in Section 4.1. As noted above, it might be worthwhile considering using a more comprehensive snow model that allows the evolution of the snowpack to be reconstructed and compared to observations. The approach to define whether a period is snow covered or not is described in Section 2.3 but from this it is hard to know just how much snow there is on the ground at any one time. Modelling the evolution of the snowpack and comparing that to observations would seem crucial given the emphasis on establishing the changes in the energetics of the snowpack over a longer temporal period.

Qg was directly measured using a Hukseflux heat flux plate at a depth of 5cm (L137) and was measured in W/m2, which was then converted to MJ/m2.

We appreciate the suggestion for a more comprehensive snow model and have decided to add a snowpack energy storage term that better accounts for non-closure of the energy balance and storage of energy in the system that may not be used immediately in melt.

While we acknowledge that a more comprehensive snow model is a valuable tool, we believe that reconstruction of the snowpack through modelling is outside the scope of this work as the goal of this research is to identify differences in snowpack energy fluxes under different synoptic regimes rather than the calculation of snowpack dynamics. This paper is focused on the bulk energy supplied to the snowpack and the changes in this at the multi-decadial scale. Snow depth information is indeed important, however the purpose of identification of homogeneous snowpack periods is not to relate the energy fluxes to quantified snowmelt. Instead, it is to assure that measurements of fluxes obtained over snowcover are directly comparable without periods of snowpack heterogeneity that would result in fluxes to bare soil or vegetation.

**R2C3:** Section 3.2; It could be useful to more carefully compare the energy balance results derived using the eddy covariance data in this study to those generated by Bilash et al. (2018) using a validated bulk aerodynamic approach. Monthly and entire cold season (winter) data could be compared prior to presenting it within a daily synoptic type framework. This suggestion stems from the fact that the presentation of the energy balance results in the current manuscript are framed differently to those described by Bilash et al. (2018), where it is stated that 80% of the energy is sourced from incoming longwave energy. It may be that the data are essentially the same but the way they are presented places emphasis on different components of the energy balance. For example, on L519 it is stated that net shortwave radiation contributes the largest amount of energy to the snowpack, which is a different message to Bilash et al. (2018).

While it appears that our results may conflict with Bilish et al. (2018), the methods and presentation of the data are the primary difference between the two papers. Bilish et al. (2018) calculated relative contributions to snowmelt by using only positive energy fluxes of Qh, Qe, L*, Qg, and Qr with K* being the only net value used. We adopt the approach of using net values of all energy balance variables as we believe that analysis of a closed system (ideally) energy balance should include both incoming and outgoing energy flux terms (Marks and Dozier, 1992; Stoy et al., 2018; Welch et al., 2016).

**Manuscript change**: Add a section to the discussion to present the data processed using a similar approach (which we have done), confirming that our results agree with Bilish et al. (2018) when the methods from the Bilish paper are used on our data.

**R2C4:** L534-L537: It is suggested that "many prior works have not made the distinction" between shortwave and longwave radiation. Historically, this may have been the case but there are numerous examples of more recent research over snow and ice that have used similar radiometers to those used by the authors to carefully identify the importance of individual radiation terms. The importance of moisture and clouds has been focused on by a number of researchers working over mid-latitude glaciers and ice sheets.

**Manuscript change**: Revise L534-L537 wording to reflect that historical works have not made that distinction, but more current works do, also see response to R2C3 – we will include a section to undertake the analysis using the newer individual term approach.

Technical and specific corrections
**R2TC1:** L132: There is a need to be more explicit that two winters are used as the observational dataset, not to just state when the observations began. L324: K and L are not explicitly defined in the manuscript in the form presented. L334: Manual classification of cloud cover requires clarification.

**Manuscript change**: Add wording to L132 to clarify that two years of data when there was a homogenous snow cover on the ground were used in the analysis.

**Manuscript change**: Change K and L on lines L324 and 325 to be K* and L*.

**Manuscript change**: Refer back to L188 when talking about classification of cloud cover on L334. Remove "manual" from cloud classification description on L334.

**References Provided by Reviewer 2:**

Bilish, S. P., McGowan, H. A., and Callow, J. N.: Energy balance and snowmelt drivers of a marginal subalpine snowpack, Hydrological Processes, 32, 3837-3851, https://doi.org/10.1002/hyp.13293, 2018.

Bilish, S. P., Callow, J. N., McGrath, G. S., and McGowan, H. A.: Spatial controls on the distribution and dynamics of a marginal snowpack in the Australian Alps, Hydrological Processes, https://doi.org/10.1002/hyp.13435, 2019.

Bormann, K. J., McCabe, M. F., and Evans, J. P.: Satellite based observations for seasonal snow cover detection and characterisation in Australia, Remote Sensing of Environment, 123, 57–71, https://doi.org/10.1016/j.rse.2012.03.003, 2012.

Cullen, N. J. and Conway, J. P.: A 22-month record of surface meteorology and energy balance from the ablation zone of Brewster Glacier, New Zealand, Journal of Glaciology, 61, 931–946, https://doi:10.3189/2015JoG15J004, 2015.

Cullen, N.J., Gibson, P.B., Mölg, T., Conway, J., Sirguey, P., and Kingston, D. G.: The influence of weather systems in controlling mass balance in the Southern Alps of New Zealand. Journal of Geophysical Research: Atmospheres, 124, https://doi:10.1029/2018JD030052, 2019.

Isaksen, K., Nordli, Ø., Førland, E. J., Łupikasza, E., Eastwood, S. and Niedz´wiedz´, T.: Recent warming on Spitsbergen - Influence of atmospheric circulation and sea ice cover. Journal of Geophysical Research: Atmospheres, 121, 11,913–11,931. https://doi:10.1002/2016JD025606, 2016

Käsmacher, O. and Schneider, C.: An objective circulation pattern classification for the region of Svalbard. Geografiska Annaler: Series A, Physical Geography, 93, 259-271. https://doi:10.1111/j.1468-0459.2011.00431.x, 2011.

Matthews, T., Hodgkins R., Wilby R. L., Guˇrmundsson S., Pálsson F., Björnsson H., & Carr S.: Conditioning temperatureâˇ Rindex model parameters on synopticˇ weather types for glacier melt simulations. Hydrological Processes, 29, 1027-1045. https://doi.org/10.1002/hyp.10217, 2015.

Romolo, L., Prowse, T.D., Blair, D., Bonsal, B.R., & Martz, L.W.: The synoptic controls on hydrology in the upper reaches of the Peace River Basin. Part I: Snow Accumulation. Hydrological Processes, 20(19), 4097–4111. https://doi.org/10.1002/hyp.6421, 2006.

Yarnal, B.: Relationships between synoptic-scale atmospheric circulation and glacier mass balance in south-western Canada during the international hydrological decade, 1965–74. Journal of Glaciology, 30, 188–198, 1984.

**Author Response References:**

Bilish, S. P., McGowan, H. A., and Callow, J. N.: Energy balance and snowmelt drivers of a marginal subalpine snowpack, Hydrol Process, 32, 3837-3851, 2018.

Breiman, L.: Random Forests, Machine Learning, 45, 5-32, 10.1023/a:1010933404324, 2001.

Chubb, T. H., Siems, S. T., and Manton, M. J.: On the Decline of Wintertime Precipitation in the Snowy Mountains of Southeastern Australia, J Hydrometeorol, 12, 1483-1497, 10.1175/Jhm-D-10-05021.1, 2011.

Gellie, N. J. H.: Native vegetation of the Southern Forests: South-east highlands, Australian alps, south-west slopes and SE corner bioregions, Royal Botanic Gardens, 2005.

Hammerle, A., Haslwanter, A., Schmitt, M., Bahn, M., Tappeiner, U., Cernusca, A., and Wohlfahrt, G.: Eddy covariance measurements of carbon dioxide, latent and sensible energy fluxes above a meadow on a mountain slope, Bound-Lay Meteorol, 122, 397-416, 10.1007/s10546-006-9109-x, 2007.

Hiller, R., Zeeman, M. J., and Eugster, W.: Eddy-covariance flux measurements in the complex terrain of an Alpine valley in Switzerland, Bound-Lay Meteorol, 127, 449-467, 10.1007/s10546-008-9267-0, 2008.

Marks, D., and Dozier, J.: Climate and Energy Exchange at the Snow Surface in the Alpine Region of the Sierra-Nevada .2. Snow Cover Energy-Balance, Water Resour Res, 28, 3043-3054, Doi 10.1029/92wr01483, 1992.

Neale, S. M., and Fitzharris, B. B.: Energy balance and synoptic climatology of a melting snowpack in the Southern Alps, New Zealand, International Journal of Climatology, 17, 1595-1609, 10.1002/(SICI)1097-0088(19971130)17:14<1595::AID-JOC213>3.0.CO;2-7, 1997.

Stiperski, I., and Rotach, M. W.: On the Measurement of Turbulence Over Complex Mountainous Terrain, Bound-Lay Meteorol, 159, 97-121, 10.1007/s10546-015-0103-z, 2016.

Stoy, P. C., Peitzsch, E., Wood, D., Rottinghaus, D., Wohlfahrt, G., Goulden, M., and Ward, H.: On the exchange of sensible and latent heat between the atmosphere and melting snow, Agricultural Forest Meteorology, 252, 167-174, 2018.

Theobald, A., McGowan, H., Speirs, J., and Callow, N.: A Synoptic Classification of Inflow-Generating Precipitation in the Snowy Mountains, Australia, J Appl Meteorol Clim, 54, 1713-1732, 10.1175/Jamc-D-14-0278.1, 2015.

Theobald, A., McGowan, H., and Speirs, J.: Trends in synoptic circulation and precipitation in the Snowy Mountains region, Australia, in the period 1958-2012, Atmos Res, 169, 434-448, 10.1016/j.atmosres.2015.05.007, 2016.

Welch, C. M., Stoy, P. C., Rains, F. A., Johnson, A. V., and McGlynn, B. L.: The impacts of mountain pine beetle disturbance on the energy balance of snow during the melt period, Hydrol Process, 30, 588-602, 10.1002/hyp.10638, 2016.